



# An improved tropospheric NO₂ column retrieval algorithm for TROPOMI over Europe

Song Liu[1], Pieter Valks[1], Gaia Pinardi[2], Jian Xu[1], Ka Lok Chan[1], Athina Argyrouli[1,3], Ronny Lutz[1], Steffen Beirle[4], Ehsan Khorsandi[5], Frank Baier[5], Vincent Huijnen[6], Alkiviadis Bais[7], Sebastian Donner[4], Steffen Dörner[4], Myrto Gratsea[8], François Hendrick[2], Dimitris Karagkiozidis[7], Kezia Lange[9], Ankie J.M. Piters[6], Julia Remmers[4], Andreas Richter[9], Michel Van Roozendael[2], Thomas Wagner[4], Mark Wenig[10], and Diego G. Loyola[1]

[1]Deutsches Zentrum für Luft- und Raumfahrt (DLR), Institut für Methodik der Fernerkundung (IMF), Oberpfaffenhofen, Germany
[2]Royal Belgian Institute for Space Aeronomy (BIRA-IASB), Brussels, Belgium
[3]Technical University of Munich (TUM), Department of Civil, Geo and Environmental Engineering, Chair of Remote Sensing Technology, Munich, Germany
[4]Max Planck Institute for Chemistry (MPI-C), Mainz, Germany
[5]Deutsches Zentrum für Luft- und Raumfahrt (DLR), German Remote Sensing Data Center (DFD), Oberpfaffenhofen, Germany
[6]Royal Netherlands Meteorological Institute (KNMI), De Bilt, the Netherlands
[7]Laboratory of Atmospheric Physics, Aristotle University of Thessaloniki (AUTH), Thessaloniki, Greece
[8]Institute for Environmental Research and Sustainable Development, National Observatory of Athens, Greece
[9]Institute of Environmental Physics (IUP-UB), University of Bremen, Bremen, Germany
[10]Meteorological Institute (MIM), Ludwig-Maximilians-Universität München (LMU), Munich, Germany

**Correspondence:** Song Liu (Song.Liu@dlr.de)

**Abstract.**

Launched in October 2017, the TROPOspheric Monitoring Instrument (TROPOMI) aboard Sentinel-5 Precursor provides the potential to monitor air quality over point sources across the globe with a spatial resolution as high as 5.5 km×3.5 km (7 km×3.5 km before 6 August 2019). The nitrogen dioxide (NO₂) retrieval algorithm for the TROPOMI instrument consists

of three steps: the spectral fitting of the slant column, the separation of stratospheric and tropospheric contributions, and the conversion of the slant column to a vertical column using an air mass factor (AMF) calculation. In this work, an improved tropospheric NO₂ retrieval algorithm from TROPOMI measurements over Europe is presented.

The stratospheric estimation is implemented using the STRatospheric Estimation Algorithm from Mainz (STREAM), which was developed as a verification algorithm for TROPOMI and does not require chemistry transport model data as input. A

directionally dependent STREAM (DSTREAM) is developed to correct for the dependency of the stratospheric NO₂ on the viewing geometry by up to $2 \times 10^{14}$ molec/cm². Applied to synthetic TROPOMI data, the uncertainty in the stratospheric column is $3.5 \times 10^{14}$ molec/cm² for polluted conditions. Applied to actual measurements, the smooth variation of stratospheric NO₂ at low latitudes is conserved, and stronger stratospheric variation at higher latitudes are captured.

For AMF calculation, the climatological surface albedo data is replaced by geometry-dependent effective Lambertian equiv-

alent reflectivity (GE_LER) obtained directly from TROPOMI measurements with a high spatial resolution. Mesoscale-





resolution a priori NO$_2$ profiles are obtained from the regional POLYPHEMUS/DLR chemistry transport model with the TNO-MACC emission inventory. Based on the latest TROPOMI operational cloud parameters, a more realistic cloud treatment is provided by a clouds-as-layers (CAL) model, which treats the clouds as uniform layers of water droplets, instead of the clouds-as-reflecting-boundaries (CRB) model, in which clouds are simplified as Lambertian reflectors.

For the error analysis, the tropospheric AMF uncertainty, which is the largest source of NO$_2$ uncertainty for polluted scenarios, ranges between 20% and 50%, leading to a total uncertainty in the tropospheric NO$_2$ column in the 30-60% range. From a validation performed with ground-based multi-axis differential optical absorption spectroscopy (MAX-DOAS) measurements, the improved tropospheric NO$_2$ data shows good correlations for nine European urban/suburban stations with an average correlation coefficient of 0.78. The implementation of the algorithm improvements leads to a decrease of the relative difference
from -55.3% to -34.7% on average.

## 1   Introduction

Tropospheric nitrogen dioxide (NO$_2$) is an important atmospheric trace gas because of its contribution to the formation of tropospheric ozone, urban haze, and acid deposition (Charlson and Ahlquist, 1969; Crutzen, 1970; McCormick, 2013). NO$_2$ is a prominent air pollutant affecting the human respiratory system (World Health Organization, 2006). Substantial amounts of
NO$_2$ are produced in the boundary layer from combustion processes. In Europe, cities regularly exceed the air quality standards for NO$_2$ (European Commission, 2017), with road transport as the largest contributor, ahead of the energy and the industry sectors (Crippa et al., 2018).

    To monitor and quantify the NO$_2$ column, NO$_2$ measurements have been provided for more than a decade on a global scale and daily basis by European satellite instruments, such as Global Ozone Monitoring Experiment (GOME) (Burrows et al.,
1999), SCanning Imaging Absorption SpectroMeter for Atmospheric CHartographY (SCIAMACHY) (Bovensmann et al., 1999), Ozone Monitoring Instrument (OMI) (Levelt et al., 2006, 2018), and Global Ozone Monitoring Experiment-2 (GOME-2) (Callies et al., 2000; Munro et al., 2006, 2016), complementary to sparse measurements by ground-based instruments. The satellite-based NO$_2$ dataset has been extended by the new generation TROPOspheric Monitoring Instrument (TROPOMI) (Veefkind et al., 2012) with an unprecedented spatial resolution of 5.5 km in the along-track direction (7 km before 6 August
2019) and 3.5 km in the across-track direction.

    TROPOMI is a nadir-viewing push-broom imaging spectrometer aboard the European Space Agency (ESA) Sentinel-5 Precursor satellite, launched on 13 October 2017. The TROPOMI instrument measures the Earth's backscattered radiance and extraterrestrial solar irradiance in the spectral range between the ultraviolet and the shortwave infrared. The spectral resolution and sampling are 0.54 and 0.20 nm in the visible channel (400 - 496 nm) used for the detection of NO$_2$. From
a sun-synchronous polar orbit, TROPOMI provides trace gas measurements as well as cloud and aerosol properties with an ascending node equatorial crossing at ∼13:30 local time. The swath width is ∼2600 km in the direction across the track of the satellite, allowing daily global coverage.



NO$_2$ measurements from TROPOMI have been widely used for ground level concentration estimates (e.g. Cooper et al., 2020; Li et al., 2020) and emission estimates (e.g. Lorente et al., 2019; Beirle et al., 2019; van der A et al., 2020; Huber et al., 2020). The high spatial resolution and good data quality allow a detailed analysis of local distribution and evolution of NO$_2$ (e.g. Stavrakou et al., 2020; Goldberg et al., 2020a; Georgoulias et al., 2020), which are particularly important and helpful during the COVID-19 pandemic (e.g. Bauwens et al., 2020; Liu et al., 2020a; Goldberg et al., 2020b; Huang and Sun, 2020; Ding et al., 2020; Koukouli et al., 2021; Biswal et al., 2020).

Independent from the operational processing, the scientific NO$_2$ retrieval algorithm for the TROPOMI instrument developed at DLR starts with the calculation of the slant column (the concentration integrated along the total light path through the atmosphere along the way from the sun to the satellite) from the TROPOMI reflectance spectra using the differential optical absorption spectroscopy (DOAS) method (Platt and Stutz, 2008). To determine the tropospheric NO$_2$ slant column, the stratospheric contribution is estimated and removed from the total slant column, after which both total and tropospheric slant columns are converted to vertical columns by the application of air mass factors (AMF).

The retrieval of tropospheric NO$_2$ columns from total column data requires an accurate stratospheric estimation, a procedure referred to as stratosphere-troposphere separation (e.g. Leue et al., 2001; Bucsela et al., 2006). One typical stratosphere-troposphere-separation algorithm is the modified reference sector method, which uses measurements over regions with negligible tropospheric NO$_2$ abundance to estimate the stratospheric NO$_2$ columns based on the assumption of longitudinally homogeneous stratospheric NO$_2$ fields. A more sophisticated approach used by the operational TROPOMI NO$_2$ retrieval relying on a chemistry transport model is data assimilation, in which the three-dimensional distributions of NO$_2$ are regularly updated such that the modelled stratospheric NO$_2$ concentrations are in close agreement with satellite measurements for low-tropospheric contributions. Compared to data assimilation, the modified reference sector method is in general simple and requires no additional model input. Therefore, the STRatospheric Estimation Algorithm from Mainz (STREAM) method (Beirle et al., 2016), which belongs to the modified reference sector methods, has been developed as a verification algorithm for TROPOMI, as a complement to the operational stratospheric correction based on data assimilation.

The quality of satellite tropospheric NO$_2$ measurements is strongly related to the tropospheric AMFs, which are determined with a radiative transfer model and depend on ancillary information such as surface albedo, vertical shape of the a priori NO$_2$ profile, cloud, and aerosol. The importance of these parameters in NO$_2$ retrievals has been recognized for OMI (e.g. Heckel et al., 2011; Lin et al., 2014; Laughner et al., 2018; Qin et al., 2019), GOME-2 (e.g. Valks et al., 2011; Lorente et al., 2018; Liu et al., 2019, 2020c), and TROPOMI (e.g. Griffin et al., 2019; Liu et al., 2020b; Zhao et al., 2020; Ialongo et al., 2020; Tack et al., 2020).

The surface albedo has implications for satellite retrievals of aerosols, clouds, and trace gases including NO$_2$. Most of current satellite NO$_2$ retrievals (Boersma et al., 2011; Liu et al., 2019, 2020c; van Geffen et al., 2020b) rely on monthly Lambertian-equivalent reflectivity (LER) climatologies derived from satellite measurements such as OMI (Kleipool et al., 2008) and GOME-2 (Tilstra et al., 2017, 2019). However, this simple assumption of isotropic surface reflection can introduce a bias by up to 35% in the NO$_2$ AMF calculation (Lorente et al., 2018). To account for the geometry-dependent surface scattering characteristics, previous works have applied measurements from the MODerate resolution Imaging Spectroradiometer



(MODIS) dataset (e.g. Vasilkov et al., 2017; Qin et al., 2019). In this study we use a new algorithm developed at DLR to re-trieve geometry-dependent effective LER (GE_LER) in the VIS based on the full-physics inverse learning machine (FP_ILM)

technique (Loyola et al., 2020b). Compared to the typical climatological LER or the directionally dependent (DLER) data (Tilstra et al., 2021), the GE_LER data represents better the actual surface conditions such as snow/ice scenarios based on each single TROPOMI measurements with a high spatial resolution. GE_LER has been successfully applied in the retrievals of TROPOMI total ozone columns in the UV (Loyola et al., 2020b) and cloud parameters in the NIR (Loyola et al., 2020a) and is being used in the corresponding operational version 2.1 cloud products (see introduction below).

The varying sensitivity of the satellite to $NO_2$ at different altitudes is considered in the tropospheric AMF calculation using vertically resolved box-AMFs and a priori $NO_2$ profiles. Typically prescribed by a chemistry transport model, the importance of applying a priori $NO_2$ profiles with sufficiently detailed resolution has been addressed (e.g. Russell et al., 2011; McLinden et al., 2014; Kuhlmann et al., 2015; Boersma et al., 2016; Laughner et al., 2016), particularly for TROPOMI with a small pixel size (Griffin et al., 2019; Liu et al., 2020b; Ialongo et al., 2020; Tack et al., 2020). Routine simulations of tropospheric trace

gases and aerosols have been provided by POLYPHEMUS/DLR since 2014 with a spatial resolution of $0.2° \times 0.3°$ (latitude, longitude) covering Europe and parts of North Africa. POLYPHEMUS/DLR is an air quality modelling platform operated at DLR based on the POLYPHEMUS chemistry transport model (Mallet et al., 2007) coupled to the Weather Research and Forecasting (WRF) model (Skamarock et al., 2008). It has been further developed within the PASODOBLE project for sensi-tivity studies of the mountainous Black Forest region (Bergemann et al., 2012) and to cover urban areas in southern Bavaria

(Khorsandi et al., 2018). It uses the TNO-MACC emission inventory (Kuenen et al., 2014). Daily model forecasts are freely available via DLR Geospatial Web Services (http://wdc.dlr.de/cgi-bin/produkt_4d_w?).

The $NO_2$ retrieval is affected by the presence of clouds, because high clouds shield underlying parts of the atmosphere, and low clouds can enhance the $NO_2$ absorption due to cloud albedo and multiple scattering if they are below or at the same height as the $NO_2$ layer (Martin et al., 2002; Kokhanovsky and Rozanov, 2008). The operational cloud retrieval for

TROPOMI is implemented using Optical Cloud Recognition Algorithm (OCRA) and Retrieval Of Cloud Information using Neural Networks (ROCINN). In addition to the retrieval product based on the assumption that clouds are simple Lambertian reflecting surfaces, referred to as Clouds-as-Reflecting-Boundaries (CRB) model, a more sophisticated set of cloud products is provided by OCRA/ROCINN, which considers clouds as optically uniform layers of scattering liquid water spherical particles, referred to as Clouds-as-Layers (CAL) model. The more realistic CAL model is regarded as the preferred method, particularly

for small TROPOMI ground pixels and for low clouds (Compernolle et al., 2020a). With an updated OCRA/ROCINN processor version 2.1 in operation since August 2020, new features such as the application of GE_LER to describe the surface albedo have been added in OCRA/ROCINN (Loyola et al., 2020a).

The satellite $NO_2$ data has been widely validated by comparison with correlative ground-based multi-axis differential optical absorption spectroscopy (MAX-DOAS) measurements (e.g. Celarier et al., 2008; Irie et al., 2008; Ma et al., 2013; Pinardi et al.,

2014, 2015; Drosoglou et al., 2017, 2018; Chan et al., 2020; Pinardi et al., 2020). MAX-DOAS measures the vertically resolved abundances of atmospheric trace species in the lowermost troposphere (Hönninger et al., 2004; Wagner et al., 2004; Wittrock et al., 2004; Heckel et al., 2005). Based on the scattered sky light under different viewing directions, high $NO_2$ sensitivity close



to the surface is obtained for the smallest elevation angles, whereas measurements at higher elevations provide information on the rest of the column.

In this paper, a number of improvements to the tropospheric $NO_2$ retrieval over Europe are introduced. To estimate and remove the stratospheric contribution, an improved STREAM algorithm is developed and evaluated by applying it to synthetic TROPOMI data and actual satellite observations. To calculate the tropospheric AMFs, the surface albedo is described by the GE_LER data consistently in both $NO_2$ and cloud retrievals; a priori $NO_2$ profiles are obtained from the regional POLYPHE-MUS/DLR chemistry transport model; the CAL cloud model from the new version 2.1 OCRA/ROCINN processor is used for

cloud correction.

     In Sect. 2, we introduce the reference algorithm at DLR for the TROPOMI $NO_2$ retrieval, which is based on an improved algorithm originally designed for GOME-2 (Liu et al., 2019) and adapted for TROPOMI measurements with optimization related to the specific instrumental aspects. In Sect. 3 and 4, we improve the stratosphere-troposphere separation and the tropospheric AMF calculation, respectively. In Sect. 5, examples of applying the retrieval algorithm to TROPOMI measurements are shown, and the error estimates are discussed. In Sect. 6, we show a comprehensive validation of the TROPOMI $NO_2$ data

using ground-based MAX-DOAS observations in Europe.

## 2    Reference retrieval for TROPOMI $NO_2$ measurement

### 2.1    DOAS slant column retrieval

Applied to the backscattered spectra measured by TROPOMI, the DOAS fit (Platt and Stutz, 2008) is a least-squares inversion

to isolate the trace gas absorption from the background processes, which are approximated by a fifth-order polynomial $P(\lambda)$ at wavelength $\lambda$:

$$\ln\left[\frac{I(\lambda)+offset(\lambda)}{I^0(\lambda)}\right] = -\sum_g S_g \sigma_g(\lambda) - \alpha_R R(\lambda) - P(\lambda). \tag{1}$$

The measurement-based term is defined as the natural logarithm of the measured earthshine radiance spectrum $I(\lambda)$ divided by the daily solar irradiance spectrum $I^0(\lambda)$. The DOAS fit is performed in the 405-465 nm wavelength range for consistency

with other $NO_2$ retrievals from TROPOMI (van Geffen et al., 2020a) and the heritage instrument OMI (van Geffen et al., 2015; Zara et al., 2018).

     The spectral effect from the absorption of a species $g$ is determined by the fitted slant column density $S_g$ and associated absorption cross-section $\sigma_g(\lambda)$:

-   $NO_2$ absorption at 220K from Vandaele et al. (2002);

-   ozone ($O_3$) absorption at 228K from Brion et al. (1998);

-   water vapor ($H_2O_{vap}$) absorption at 293K from Rothman et al. (2010), rescaled as in Lampel et al. (2015);

-   oxygen dimer ($O_4$) absorption at 293K from Thalman and Volkamer (2013);





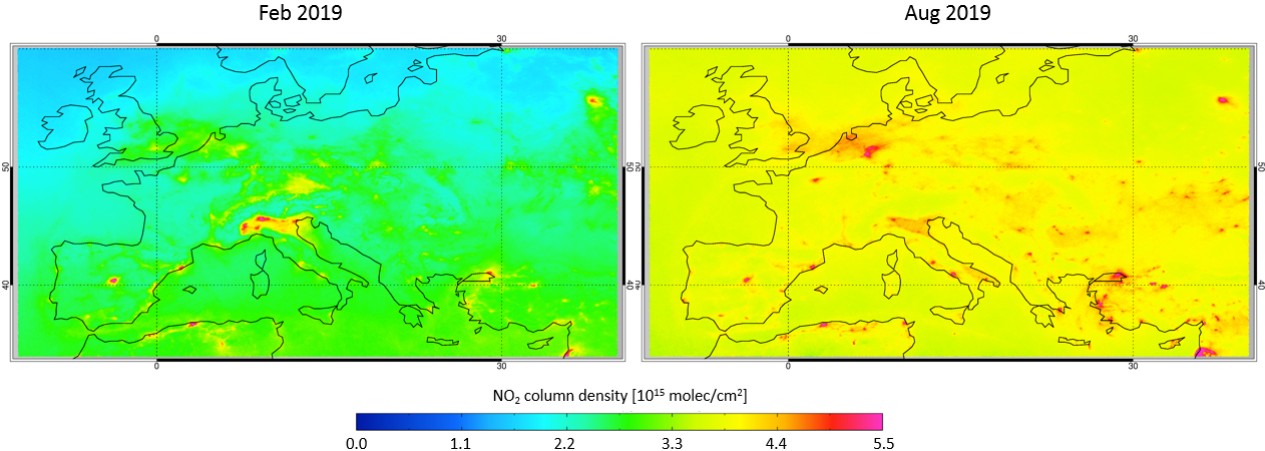

**Figure 1.** TROPOMI NO$_2$ slant columns (scaled by geometric AMFs) over Europe in February and August 2019.

– liquid water (H$_2$O$_{liq}$) absorption at 297K from Pope and Fry (1997), smoothed as in Peters et al. (2014).

The contribution of the rotational Raman scattering to the measured spectrum, namely the Ring effect (Grainger and Ring, 1962; Solomon et al., 1987), is treated as a pseudo absorber, by means of an additive Ring reference spectrum $R(\lambda)$ and a scaling coefficient $\alpha_R$ as fitting parameter. A linear intensity offset correction $offset(\lambda)$ is fitted as an additional effective cross-section to correct for the stray light in the spectrometer, the inelastic scattering in the ocean, and remaining calibration issues in the level 1 data (Liu et al., 2019). The TROPOMI level 1b version 1 spectra are analysed using the QDOAS software developed at BIRA-IASB (Fayt et al., 2011; Danckaert et al., 2017). Figure 1 shows examples of the TROPOMI NO$_2$ slant columns over Europe in February and August 2019, where large NO$_2$ hotspots can be identified. Note that the slant columns are scaled by geometric AMFs to correct for the angular dependencies of TROPOMI measurements.

The NO$_2$ slant columns from single orbits show an across-track striping pattern, a well-known feature of observations of push-broom spectrometers such as OMI (Boersma et al., 2011) and TROPOMI (van Geffen et al., 2020a), which is likely caused by the viewing zenith angle (VZA) dependency of the spectral calibration and detector sensitivity (Boersma et al., 2018). To reduce the systematic stripes, a de-striping correction amplitude is calculated empirically (Boersma et al., 2011) based on the daily averaged across-track variability of NO$_2$ slant columns over clean regions between 20°S and 20°N. The magnitude of the NO$_2$ de-striping correction is up to $1 \times 10^{14}$ molec/cm$^2$ and is stable over time (not shown), in agreement with the operational TROPOMI de-striping that relies on the chemistry transport model data (van Geffen et al., 2020a).

## 2.2 Stratosphere-troposphere separation

The stratospheric NO$_2$ component is estimated using the STREAM method (Beirle et al., 2016). Belonging to the modified reference sector methods, STREAM uses total NO$_2$ column measurements over clean and remote regions as well as over clouded scenes with negligible tropospheric columns. STREAM calculates weighting factors for each satellite pixel to define the con-



tribution of total columns to the stratospheric estimation: potentially polluted pixels are weighted low, cloudy observations with medium cloud heights are weighted high, and the weights are further adjusted in case of large biases in the tropospheric

residues. Depending on these weighting factors, stratospheric $NO_2$ fields are derived by a weighted convolution of the total columns using convolution kernels, which are wider at lower latitudes to account for the low longitudinal variability assumption of stratospheric $NO_2$ and narrower at higher latitudes to reflect the stronger natural variations.

STREAM was developed as a verification algorithm for the TROPOMI instrument, as a complement to the operational stratospheric correction based on data assimilation (van Geffen et al., 2020b). STREAM has been successfully applied to the

$NO_2$ measurements from GOME, SCIAMACHY, OMI, and GOME-2 (Beirle et al., 2016; Liu et al., 2019) with the advantage of requiring no model input. In contrast to previous modified reference sector methods which normally apply a conservative masking approach (flagging pixels as either clean or polluted and skipping the latter for stratospheric estimation) and hardly use information over continents, STREAM introduces an improved treatment of polluted and cloudy pixels by defining weighting factors for each satellite pixel. Stratospheric $NO_2$ columns from STREAM differ by up to $3 \times 10^{14}$ molec/cm$^2$ as compared

to results from data assimilation and other modified reference sector methods, within the general uncertainties of stratosphere-troposphere separation (Beirle et al., 2016; Boersma et al., 2018). The STREAM stratospheric $NO_2$ columns show an average bias of $1 \times 10^{13}$ molec/cm$^2$ with respect to the ground-based zenith-scattered light differential optical absorption spectroscopy (ZSL-DOAS) measurements (Compernolle et al., 2020b).

## 2.3 AMF calculation

The conversion between the slant column $S$ and the vertical column $V$ is implemented by division with an AMF $M$:

$$V = \frac{S}{M}. \tag{2}$$

Given the small optical depth of $NO_2$, $M$ can be derived as:

$$M = \frac{\sum_l m_l(\boldsymbol{b}) x_l c_l}{\sum_l x_l}, \tag{3}$$

where $m_l$ is the box-AMFs in layer $l$, $x_l$ is the partial column from the a priori $NO_2$ profile, and $c_l$ is a correction coefficient

to correct for the temperature dependency of the $NO_2$ cross section (Boersma et al., 2004; Nüß et al., 2006; Bucsela et al., 2013). $m_l$ is a function of model inputs $\boldsymbol{b}$, which include TROPOMI measurement geometries, surface albedo, and surface pressure. The box-AMFs $m_l$ values are calculated at 437.5 nm (near the mid-point wavelength of fitting window 405-465 nm), as recommended by Boersma et al. (2018), using the linearised vector code VLIDORT (Spurr, 2006). The light path in the troposphere is affected by scattering on air molecules as well as cloud and aerosol particles, and therefore the tropospheric

AMF calculation depends on surface albedo, a priori $NO_2$ profiles, and cloud properties. Table 1 summarises the parameters used in the AMF calculation. Figure 2 shows the tropospheric $NO_2$ columns retrieved from the reference algorithm over Europe in February and August 2019. A large amount of $NO_2$ is located in the troposphere for industrialised and urbanised areas (see Fig. 1).





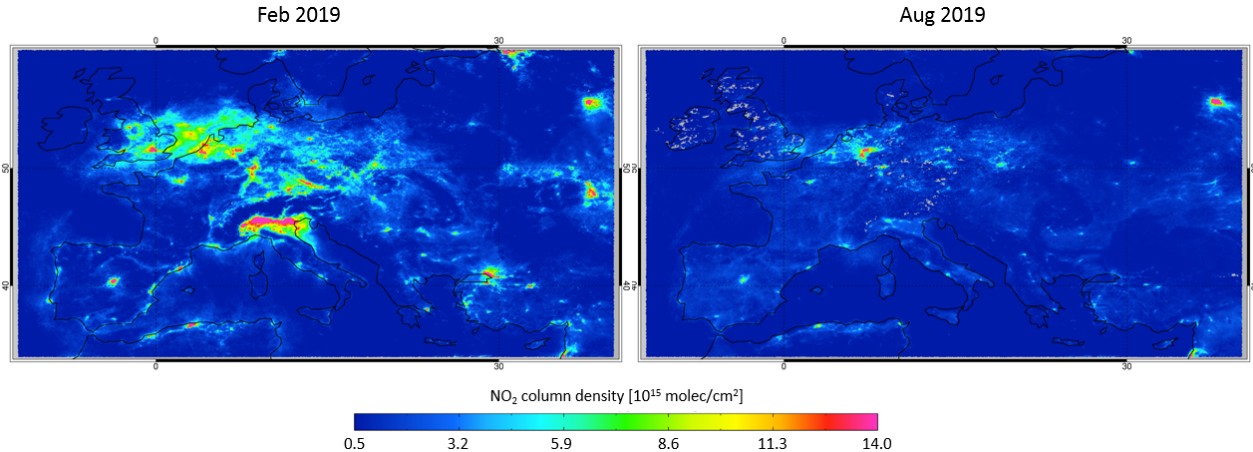

**Figure 2.** Tropospheric NO$_2$ columns from the reference algorithm over Europe in February and August 2019. Only measurements with cloud radiance fraction less than 0.5 are included.

**Table 1.** Parameters used to calculate tropospheric AMFs. See Table 2 for details of the chemistry transport models used to obtain the a priori NO$_2$ profiles.

|  | Reference retrieval | Improved algorithm |
| --- | --- | --- |
| Surface albedo | OMI LER climatology | TROPOMI GE_LER data |
| A priori NO$_2$ profile | TM5-MP | POLYPHEMUS/DLR |
| Cloud parameter | OCRA/ROCINN_CRB version 1.x | OCRA/ROCINN_CAL version 2.1 |

In the reference algorithm, the surface albedo is described by a monthly climatology based on four years (2004-2007) of OMI LER measurements at 440 nm (Kleipool et al., 2008) with a similar overpass time and viewing conditions as TROPOMI. The surface albedo for each TROPOMI pixel is calculated by an area-weighted tessellation of the OMI monthly averaged surface albedo maps (0.5°×0.5°) and a linear interpolation in time to the measurement day.

Daily TM5-MP vertical NO$_2$ profiles (Williams et al., 2017) simulated at a global 1°×1° resolution are used as a priori NO$_2$ vertical profiles due to the operational advantage, as summarized in Table 2. The a priori profiles are determined for the satellite overpass time and interpolated to the center of the TROPOMI pixel based on four nearest neighbour TM5-MP cell centers.

In the presence of clouds, the AMF calculation adopts the independent pixel approximation (Cahalan et al., 1994):

$$M = \omega M^{cl} + (1 - \omega)M^{cr}, \qquad (4)$$

where $M^{cl}$ represents the AMF for completely cloudy sky and $M^{cr}$ for completely clear sky. $M^{cl}$ and $M^{cr}$ are derived with Eq. (3) with $M^{cl}$ mainly relying on the cloud pressure (height) and the cloud albedo (optical depth). The cloud radiance fraction



**Table 2.** Summary of the chemistry transport model specifications.

|  | TM5-MP | POLYPHEMUS/DLR |
|---|---|---|
|  | (Huijnen et al., 2010; Williams et al., 2017) | (Mallet et al., 2007; Bergemann et al., 2012; Khorsandi et al., 2018) |
| Spatial resolution | $1° \times 1°$ (latitude, longitude) | $0.2° \times 0.3°$ (latitude, longitude) |
| Vertical resolution (>150 hPa) | ~18 layers | 20 layers |
| Meteorological fields | ECMWF 3 h | WRF 1 h |
| Tropospheric chemistry | Modified CB05 (Williams et al., 2013) | RACM for trace gases (Stockwell et al., 1997) SORGAM-SIREAM for aerosols (Debry et al., 2007; Schell et al., 2001) |
| Anthropogenic emission | MACCity (Granier et al., 2011) | TNO-MACC (Denier van der Gon et al., 2010; Kuenen et al., 2014) |
| Advection | Slopes scheme (Russell and Lerner, 1981) | third-order direct space-time scheme with a Koren-Sweby flux limiter |
| Convection | ECMWF | WRF (Skamarock et al., 2008) |
| Diffusion | Holtslag and Boville (1993) | second-order Rosenbrock method (Verwer et al., 2002) |

$\omega$ is derived from the TROPOMI cloud fraction $c_f$:

$$\omega = \frac{c_f I^{cl}}{(1 - c_f)I^{cr} + c_f I^{cl}} \tag{5}$$

with $I^{cl}$ and $I^{cr}$ representing the radiances for cloudy and clear scenes, respectively. $I^{cl}$ and $I^{cr}$ are calculated using the LIDORT model (Spurr et al., 2001), depending mostly on TROPOMI viewing geometries, surface albedo, and cloud albedo.

The operational TROPOMI cloud parameters from the OCRA/ROCINN algorithms (Lutz et al., 2016; Loyola et al., 2018)
with clouds treated as opaque Lambertian surfaces (CRB model) are applied for the cloud correction. OCRA derives the cloud fraction by separating a spectral scene into cloudy contribution and cloud-free background. With the OCRA cloud fraction and the surface albedo from the MERIS black-sky climatology (Popp et al., 2011) as inputs, ROCINN calculates the cloud pressure and cloud albedo by comparing the measured and simulated sun-normalised radiances in and around the $O_2$ A-band in the NIR.

The original OCRA takes the spectral information from the UV-VIS-NIR part (320 – 800 nm) and transforms the radiances
of three predefined spectral ranges to three-color reflectances (RGB: red, green, and blue region of the spectrum). The cloud-free background maps are calculated for each of these three colors. For the TROPOMI application, the OCRA color space approach is applied with two colors (GB) using the UV-VIS (350 – 495 nm) spectra to avoid the spatial misalignment between the UV-VIS and NIR footprints.

Based on the Lambertian cloud assumption, OCRA/ROCINN_CRB tends to retrieve a cloud height (at the optical centroid
of the cloud rather than the cloud top) close to the surface altitude for low cloud fraction (Compernolle et al., 2020a). The Lambertian cloud assumption is also applied in the Fast Retrieval Scheme for Clouds from the $O_2$ A-band (FRESCO) algorithm





(Koelemeijer et al., 2001; Wang et al., 2008). FRESCO for Sentinel (FRESCO-S) (Wang and Sneep, 2019) is implemented as a support product for the TROPOMI operational $NO_2$ processing (van Geffen et al., 2020b). FRESCO-S retrieves the cloud fraction and cloud pressure from the reflectance in and around the $O_2$ A-band. The cloud albedo is assumed to be 0.8, as
opposed to OCRA/ROCINN, where cloud albedo is retrieved.

## 3   New Stratosphere-troposphere separation

STREAM was originally designed for TROPOMI and optimized for OMI within TROPOMI verification activities (Beirle et al., 2016). STREAM consists basically of two steps: the definition of weighting factors for each satellite pixel and the application of weighted convolution. To identify potentially polluted areas, a climatology of tropospheric $NO_2$ columns is derived in this
study using TROPOMI $NO_2$ measurements from 2018-2019, instead of using SCIAMACHY $NO_2$ measurements from 2003-2011 as in the original STREAM. Based on the pollution weight, as well as the cloud weight and tropospheric residue weight, STREAM estimates stratospheric fields for individual orbits using a weighted convolution on $0.5° \times 0.5°$ grid pixels.

As a result of wide swath (∼2600 km), local time differences across a TROPOMI swath are considerable at high latitudes, and the $NO_2$ measurements show dependency on VZA (directly related to local time) due to the diurnal variation of strato-
spheric $NO_2$ (Dirksen et al., 2011; Belmonte Rivas et al., 2014). Figure 3 shows the total $NO_2$ columns measured by TROPOMI in January 2019 for different latitudes as a function of VZA. The impact of local time changes across the orbit is up to $2 \times 10^{14}$ molec/cm$^2$ at the swath edge for latitudes higher than $50°$, in agreement with estimations for OMI measurements (Beirle et al., 2016).

In the following, the concept of a directionally dependent STREAM (DSTREAM) is introduced to estimate the stratospheric
$NO_2$ column (Sect. 3.1). The performance of STREAM and DSTREAM is analyzed using synthetic TROPOMI $NO_2$ data (Sect. 3.2), and both algorithms are applied to TROPOMI measurements (Sect. 3.3).

### 3.1   DSTREAM

To correct for the VZA dependency of stratospheric $NO_2$, the DSTREAM is developed, which divides the orbit swath into western (VZA from ∼-66° to ∼-30°), central (VZA from ∼-30° to ∼30°), and eastern (VZA from ∼30° to ∼66°) segments.
Note that the VZA is defined negative for observations on the west side of the orbit swath throughout the study. For each of the orbit swath containers, the original STREAM is applied based on data from the respective orbit swath segment.

For each individual satellite pixel with a VZA, a directionally dependent stratospheric $NO_2$ column $V_s^{dir}$ is parameterized using a linear interpolation on the DSTREAM grid results estimated using the eastern, central, and western segments of the orbit swath. As the VZA dependency is negligible for low latitudes (from Fig. 3), and the interpolation error may increase for
$V_s^{dir}$ due to less orbital overlap, the final stratospheric $NO_2$ column $V_s$ is calculated as the weighted mean in dependence on latitude $\theta$:

$$V_s = cos^2(\theta)V_s^{ori} + sin^2(\theta)V_s^{dir}. \tag{6}$$

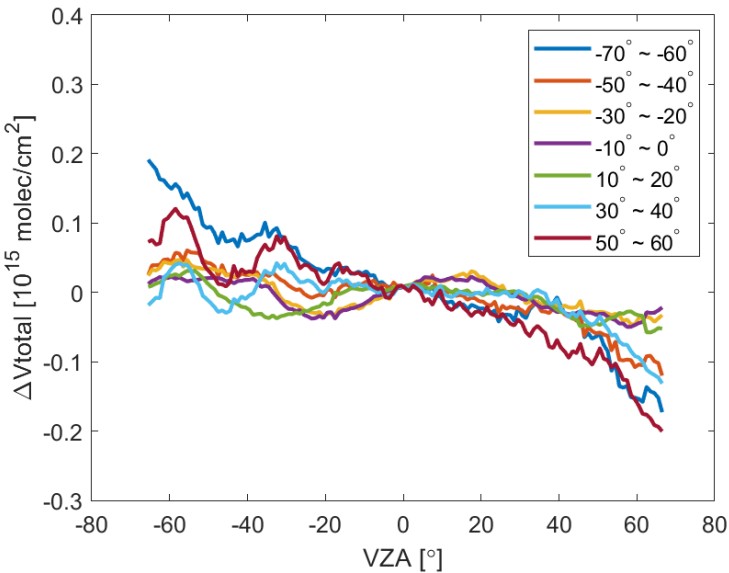

**Figure 3.** VZA dependency of TROPOMI total $NO_2$ columns (compared to nadir) at different latitudes in January 2019. The VZA is defined negative for observations on the west side of the orbit swath. The de-striping is not implemented here.

By this method, the stratospheric $NO_2$ from the original STREAM $V_s^{ori}$ is applied for equator, and the VZA dependency is captured for polar regions with significant orbital overlap.

### 3.2 Application to synthetic data

The performance of the original STREAM and the improved DSTREAM for TROPOMI is evaluated with simulated $NO_2$ fields from the IFS(CB05BASCOE) experiment (Huijnen et al., 2016). The IFS(CB05BASCOE) model is particularly advantageous for stratospheric studies due to the extension of the tropospheric chemistry module in the Integrated Forecast System (IFS) with the stratospheric chemistry from the Belgian Assimilation System for Chemical ObsErvations (BASCOE) system. The STREAM and DSTREAM are applied to the synthetic TROPOMI total $NO_2$ columns, and the estimated stratospheric $NO_2$ columns are compared with the a priori truth (stratospheric fields from model). See Liu et al. (2019) for more details on constructing synthetic total and stratospheric $NO_2$ columns using IFS(CB05BASCOE) data.

Figure 4 displays the synthetic total $NO_2$ columns from IFS(CB05BASCOE), the modelled stratospheric columns, and the estimated stratospheric columns from STREAM and DSTREAM on 1 January and 1 August 2019. The overall latitudinal and seasonal dependencies are reflected in the stratospheric fields from STREAM and DSTREAM. Smaller structures in the synthetic total columns and the modelled stratospheric columns at high latitudes, caused by the diurnal variation of stratospheric $NO_2$ across the orbital swath, are aliased into the troposphere by STREAM but captured by DSTREAM. The average difference



**Figure 4.** Synthetic total $NO_2$ columns, a priori stratospheric $NO_2$ columns from IFS(CB05BASCOE), and estimated stratospheric $NO_2$ columns from STREAM and DSTREAM on 1 February and 1 August 2019.

between the estimated and a priori results is $4 \times 10^{14}$ molec/cm² for STREAM and $3.5 \times 10^{14}$ molec/cm² for DSTREAM with improvements mainly for latitudes higher than $50°$.


**Figure 5.** Total NO$_2$ columns and stratospheric NO$_2$ columns estimated using STREAM and DSTREAM, as measured by TROPOMI on 1 February and 1 August 2019.

## 3.3 Application to TROPOMI measurements

Applying STREAM and DSTREAM to TROPOMI data, Fig. 5 shows the total columns from TROPOMI and the estimated stratospheric fields on 1 February and 1 August 2019. For both months, the STREAM and DSTREAM show similar global patterns of stratospheric NO$_2$. The stratospheric and tropospheric contributions over polluted regions are successfully separated due to the use of clean and cloudy measurements at the same latitude where the tropospheric column is shielded. The smooth background at low latitudes is conserved, and the stronger variations of stratospheric NO$_2$ at higher latitudes are captured, e.g. in the polar vortex on 1 February.




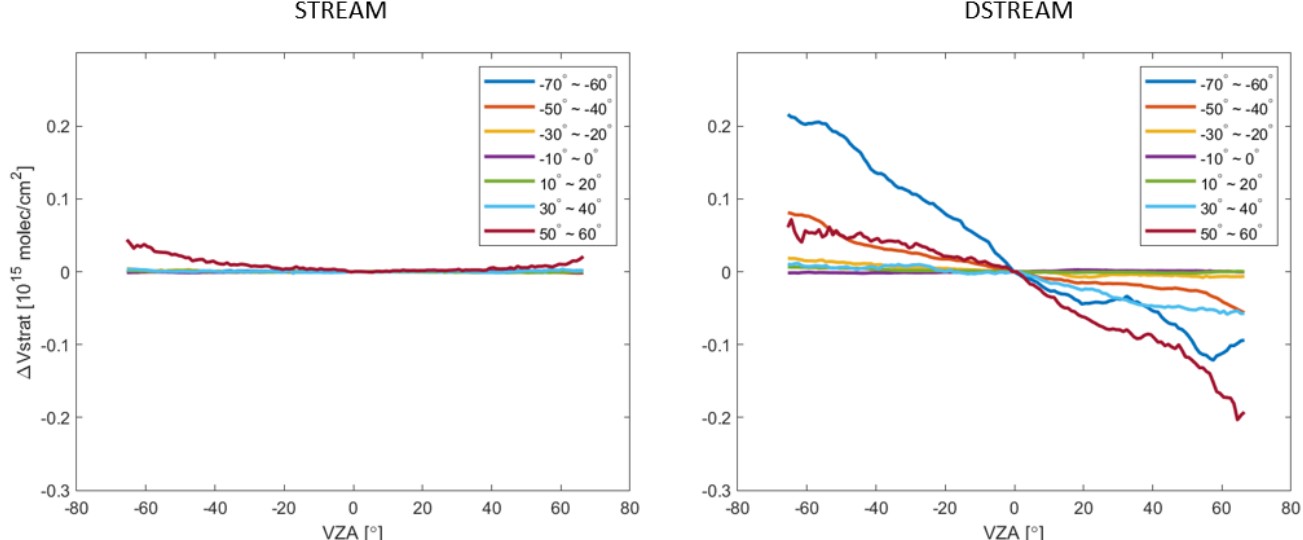

**Figure 6.** VZA dependency of TROPOMI stratospheric NO$_2$ columns estimated using STREAM and DSTREAM (compared to nadir) at different latitudes in January 2019. The VZA is defined negative for observations on the west side of the orbit swath.

Figure 6 shows the TROPOMI stratospheric columns estimated using STREAM and DSTREAM for different latitudes as a function of VZA in January 2019. The VZA dependency by up to $2 \times 10^{14}$ molec/cm$^2$ caused by the local time difference in Fig. 3 is captured by DSTREAM, particularly for high latitudes. The overestimation on the west side of swath edge and the underestimation on the east side in the STREAM results are improved in DSTREAM.

## 4 Improved AMF calculation

### 4.1 Surface albedo

Surface albedo is an important parameter for an accurate retrieval of trace gas columns and cloud properties. The sensitivity of backscattered radiance to the boundary layer NO$_2$ is strongly related to the surface albedo, especially over polluted areas. In this study, the surface albedo is described using GE_LER retrieved by the FP_ILM algorithm (Loyola et al., 2020b). Unlike conventional approaches (Rodgers, 2000; Doicu et al., 2010), FP_ILM is a machine learning based approach consisting of a training phase wherein an inverse function is derived from synthetic data generated with a radiative transfer model and an operational phase wherein the inverse function is applied to measured spectra. The FP_ILM algorithm has been employed to retrieve ozone profile shapes and sulfur dioxide layer heights from GOME-2 and TROPOMI (Xu et al., 2017; Efremenko et al., 2017; Hedelt et al., 2019).

Combining the DOAS equation Eq. (1) and the conventional forward model, our forward problem can be formulated as an approximation of the DOAS-fitted slant column density ($SCD$) and the DOAS polynomial ($P$) using the forward model ($F$)



with the solar/satellite viewing geometry ($\Theta$), effective surface pressure ($p_e$), and surface albedo ($A_s$):

$$\{SCD, P\} = F(\Theta, p_e, A_s). \tag{7}$$

300 During the training phase, synthetic TROPOMI spectra in the 405 - 465 nm range are simulated by LIDORT (Spurr et al., 2001) in conjunction with the smart sampling technique (Loyola et al., 2016). The cloud impact is considered in the simulations with the use of the effective surface pressure $p_e$, which depends on OCRA cloud fraction, ROCINN_CRB cloud pressure, and surface pressure (Loyola et al., 2020b). The aerosol influence is not considered. The DOAS fitting is applied to the simulated spectra using the consistent DOAS settings as introduced in Sect. 2.1. The simulation results from Eq. (7) are grouped as inputs to a 305 multi-layer neural network, and the neural network is trained to learn the inverse function. In the operational phase, GE_LER is generated using the trained neural network and the DOAS results from the measured spectra. An additional polynomial fitting is subsequently included to account for the bidirectional reflectance distribution function (BRDF) effect.

For consistency with the NO$_2$ retrieval, the GE_LER retrieval is performed for every single ground pixel using the same TROPOMI spectrum and DOAS configurations. Global maps are generated from the GE_LER retrievals under clear-sky con- 310 ditions (OCRA cloud fraction small than 0.05) and updated on a daily basis on a time window between one and four weeks depending on cloudiness. In contrast to the OMI LER climatology, the GE_LER data relies on the measurements from the TROPOMI instrument itself with an improved spatial resolution ($0.1°\times0.1°$) and better characterizes the actual surface conditions, particularly for snow/ice scenarios.

Figure 7 compares the climatological OMI LER data and GE_LER data for February and August 2019. The surface LER 315 values from GE_LER are lower than the climatological OMI values by 0.03 on average. The improved spatial resolution for GE_LER enables a better representation of surface features. Larger differences by more than 0.2 are found in winter over snow/ice regions such as Russia and the Alps, because GE_LER captures the actual snow/ice conditions. The GE_LER values are higher by up to 0.05 over the North Sea, due to the use of only one month of TROPOMI data compared to the multiple years for OMI climatology, which makes GE_LER more likely affected by aerosol contamination. In the near future, an improved 320 aerosol screening based on TROPOMI aerosol index data will be implemented in the GE_LER algorithm.

Figure 8 shows the monthly average differences in the tropospheric NO$_2$ columns retrieved using the climatological OMI LER and the TROPOMI GE_LER in February and August 2019. An effect is noticed mainly in winter under polluted conditions. Consistent with the LER changes in Fig. 7, the general reduced surface LER from GE_LER results in a decrease in the tropospheric AMF and thus an increase in the calculated tropospheric NO$_2$ column by up to $3 \times 10^{15}$ molec/cm$^2$. A reduction 325 by up to $1 \times 10^{15}$ molec/cm$^2$ is found for snow/ice coverages and aerosol scenes.

### 4.2 A priori NO$_2$ profiles

To account for the varying sensitivity of the satellite to NO$_2$ at different altitudes, the POLYPHEMUS/DLR simulations (Mallet et al., 2007) with a spatial resolution of $0.2°\times0.3°$ (latitude, longitude) and a temporal resolution of 1 h are applied for Europe in this study. Compared to the reference algorithm using TM5-MP a priori NO$_2$ profiles, it can be expected that the improved





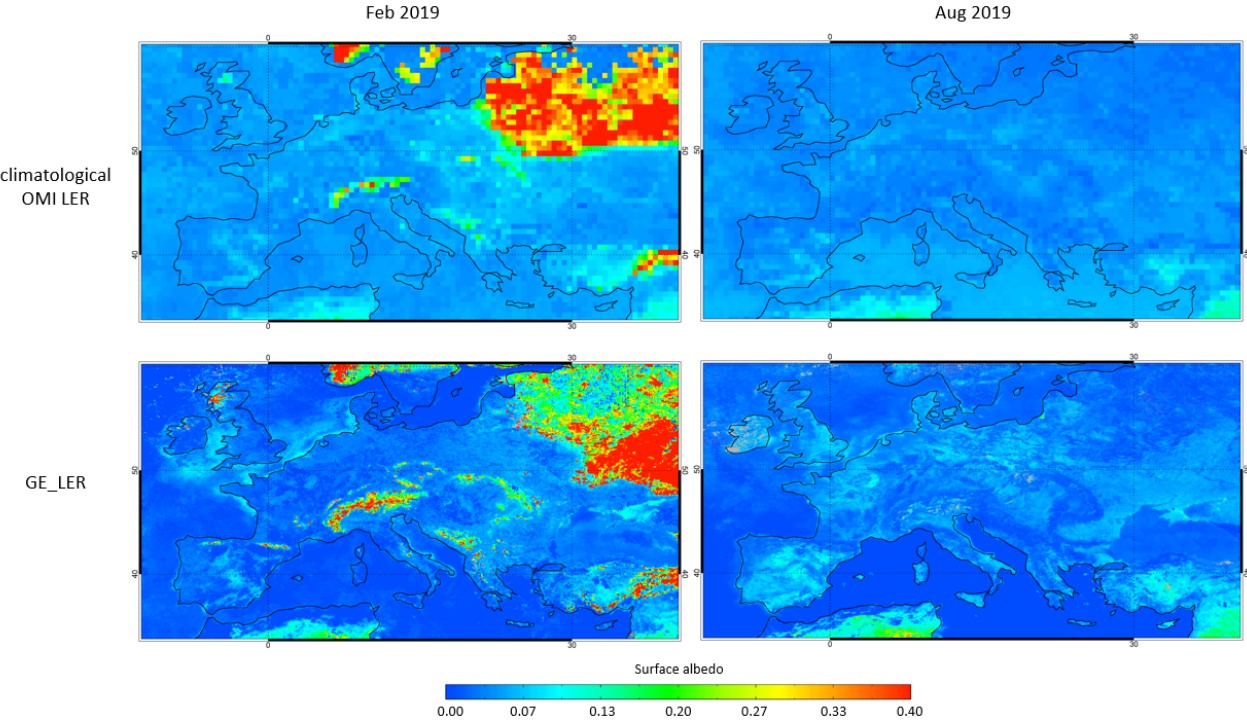

**Figure 7.** OMI surface LER climatology at 440 nm (Kleipool et al., 2008) and GE_LER retrieved from TROPOMI data in the NO$_2$ retrieval window (405 - 465 nm) over Europe in February and August 2019.

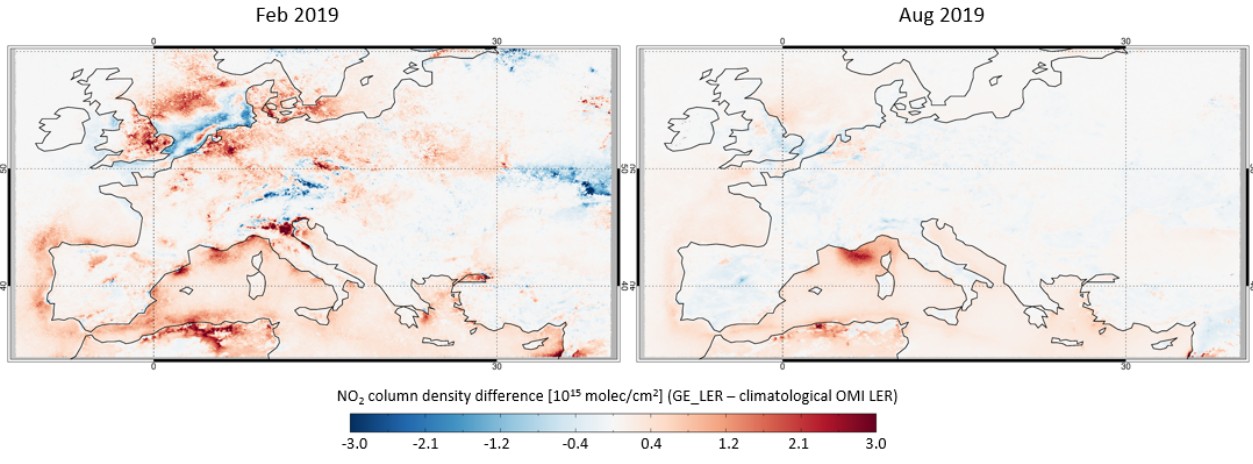

**Figure 8.** Differences in the tropospheric NO$_2$ columns retrieved using the climatological OMI LER and GE_LER over Europe in February and August 2019. Only measurements with cloud radiance fraction less than 0.5 are included.





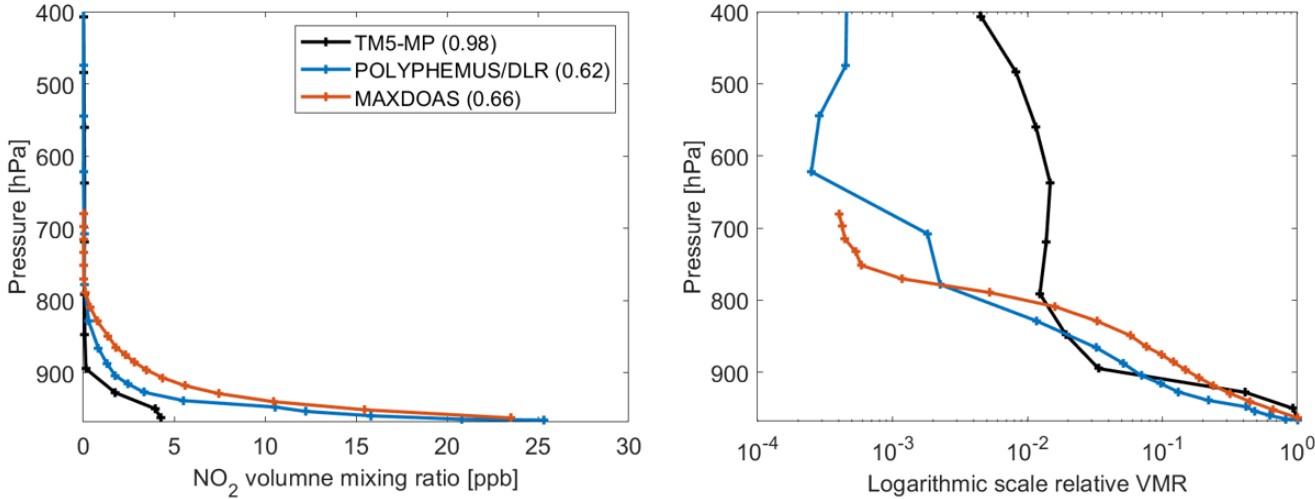

**Figure 9.** A priori $NO_2$ profiles from the chemistry transport models TM5-MP and POLYPHEMUS/DLR and the low-tropospheric $NO_2$ profile derived from the ground-based MAX-DOAS measurements over Munich in Germany (48.09°N, 11.40°E) on 5 February 2019. The calculated clear-sky tropospheric AMF is given in the bracket next to each label in the legend. Normalized profiles (to the lowest values) are also shown on a logarithm scale.

resolution of POLYPHEMUS/DLR is better able to capture accurately the local $NO_2$ distribution, particularly for regions with large heterogeneity and variability.

As summarized in Table 2, the meteorological parameters are provided by Weather Research and Forecasting (WRF) Version 3.5 daily forecasts with a 30 km×30 km spatial resolution, initialized by daily Global Forecast System (GSF) global forecast from National Oceanic and Atmospheric Administration (NOAA). The original POLYPHEMUS model is an assembly of sev-

eral Eulerian and Gaussian models for handling passive tracers, photochemistry, and aerosol dynamics (Mallet et al., 2007) and is developed based on the chemistry transport model Polair3D (Boutahar et al., 2004). In this study, the Regional Atmospheric Chemistry Modeling (RACM) chemical mechanism (Stockwell et al., 1997) is applied along with the Size-REsolved Aerosol Model (SIREAM) and Secondary ORGanic Aerosol Model (SORGAM) size-resolved aerosol model (Debry et al., 2007; Schell et al., 2001). The anthropogenic emissions are extracted from the European TNO-MACC emission inventory

(Denier van der Gon et al., 2010; Kuenen et al., 2014). With a spatial resolution of 7 km×7 km, TNO-MACC defines 10 source categories including road transport and shipping and uses source sector-specific data in a harmonized way. Biogenic emissions from soils are computed as proposed in Simpson et al. (1999). Lightning emissions are not considered.

Figure 9 shows the TM5-MP and POLYPHEMUS/DLR a priori $NO_2$ profiles over Munich in Germany (48.09°N, 11.40°E) on 5 February 2019, with the calculated clear-sky tropospheric AMFs also reported. POLYPHEMUS/DLR shows a higher

surface layer $NO_2$ concentration and yields a tropospheric AMF that is reduced by 0.36 (36.7%). Figure 9 additionally shows the low-tropospheric $NO_2$ profile derived from the ground-based MAX-DOAS data (Chan et al., 2020) and the tropospheric AMF calculated using the MAX-DOAS $NO_2$ profile as a priori information (assuming a constant profile shape for the high



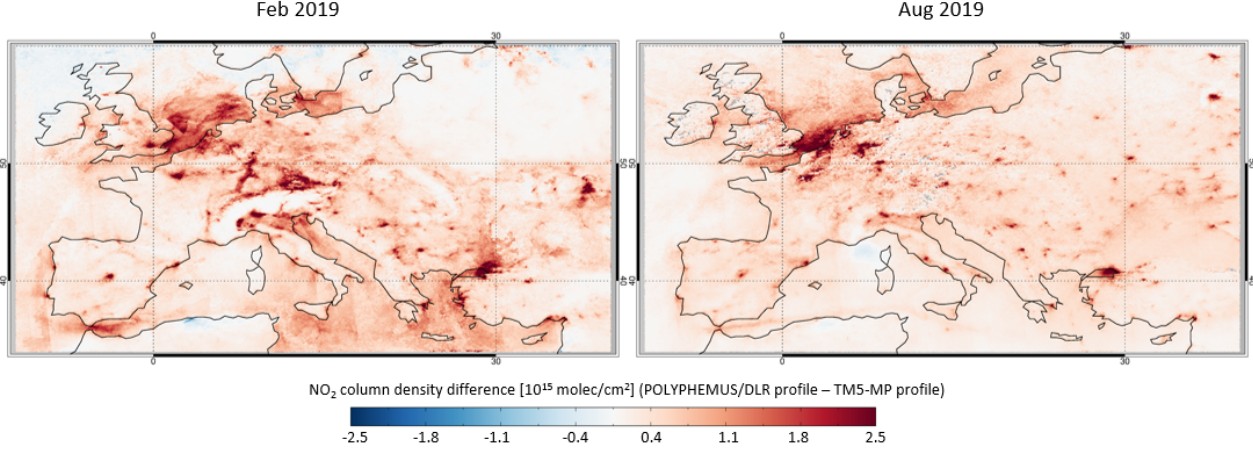

**Figure 10.** Differences in the tropospheric $NO_2$ columns retrieved using the TM5-MP and POLYPHEMUS/DLR a priori $NO_2$ profiles over Europe in February and August 2019. Only measurements with cloud radiance fraction less than 0.5 are included.

troposphere). With a typical horizontal sensitivity of a few kilometers, the MAX-DOAS profile shows large amounts of $NO_2$ in the lower troposphere (Irie et al., 2011; Wagner et al., 2011). The profile shape from POLYPHEMUS/DLR agree better
with the MAX-DOAS measurements than TM5-MP, with the tropospheric AMF bias improving from 0.32 (48.5%) to -0.04 (-6.0%).

Figure 10 shows the monthly average differences in the tropospheric $NO_2$ columns retrieved using TM5-MP and POLYPHE-MUS/DLR a priori $NO_2$ profiles in February and August 2019. POLYPHEMUS/DLR uses the TNO-MACC emission database, which generally shows higher total $NO_2$ emissions at much higher spatial resolution in comparison with the global MACCity
inventory. Further tests to investigate the sensitivity of model $NO_2$ profiles have concluded that the horizontal resolution and the representation of the tropospheric boundary layer have the largest influence (not shown). The generally steeper profile shape from POLYPHEMUS/DLR (see Fig. 9) increases the retrieved tropospheric $NO_2$ columns by more than $2 \times 10^{15}$ molec/cm$^2$ for pollution hot spots, e.g. regions with large population or heavy industry in the Benelux, northern Italy, and western Turkey, as well as highways with intense road traffic in northern Spain, southern France, and western Germany.

**4.3 Cloud correction**

**4.3.1 New OCRA/ROCINN processor version**

With a new version 2.1 OCRA/ROCINN processor in operation since August 2020 (Loyola et al., 2020a), the background maps used in the OCRA cloud fraction determination are calculated based on one year of TROPOMI data (April 2018 - March 2019) instead of the previously used OMI measurements, and the spatial resolution improves from $0.2° \times 0.4°$ to $0.1° \times 0.1°$. In
the pre-processing step, a TROPOMI-based VZA dependency correction is applied instead of using the OMI measurements.



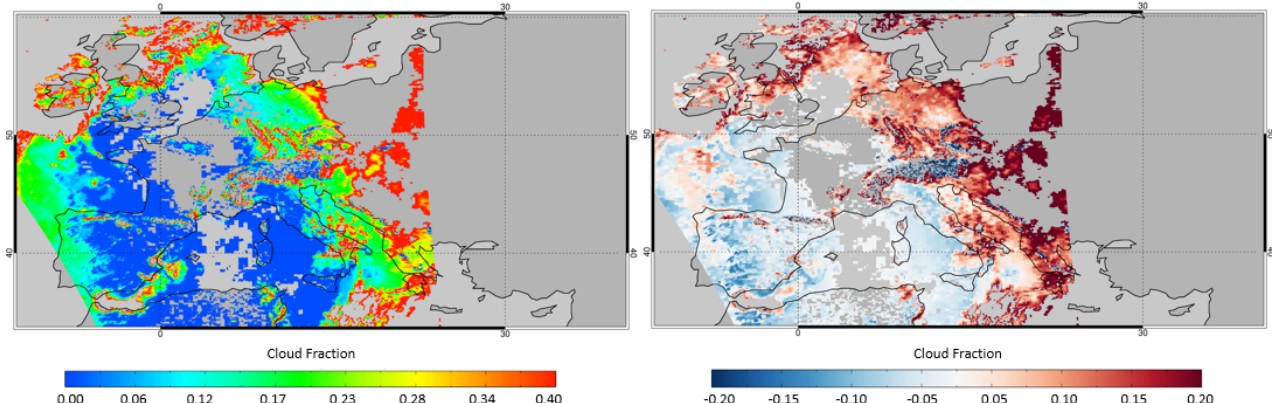

**Figure 11.** OCRA cloud fractions from the version 2.1 processor (left) and comparisons with version 1.x (right) over Europe for orbit 6939 on 14 February 2019. Only measurements with $0 <$ cloud fraction $<= 0.4$ are included.

For ROCINN, the static MERIS surface albedo climatology (Popp et al., 2011) is replaced with the dynamic on-line GE_LER retrieval in the ROCINN fitting window, which derives the surface properties directly from TROPOMI itself on a daily basis (see Sect. 4.1). The co-registration between UV-VIS band (used by OCRA) and NIR band (used by ROCINN) is optimized with a look-up table containing the fraction of overlapping area between the target and source pixels, and the cloud properties

are provided in both spectral bands. Over snow/ice surfaces indicated by the European Centre for Medium Range Weather Forecast (ECMWF) dataset, ROCINN retrieves effective cloud pressure and cloud albedo values assuming a cloud fraction of 0. These values are applied in the $NO_2$ AMF calculation when the difference between the scene pressure and the surface pressure is less than 2% and the observation is considered to be nearly cloud-free (van der A et al., 2020).

Figure 11 compares the OCRA cloud fractions from the version 1.x and 2.1 processors for orbit 6939 on 14 February 2019.

The update of cloud-free background maps increases the cloud fractions by more than 0.1 for large cloud fraction values and reduces the values by more than 0.1 for snow/ice covers, e.g. over the Alps and the Ore Mountains. Figure 12 shows the OCRA cloud fractions as a function of VZA. Mainly due to the improved VZA correction, the overestimation of cloud fractions, particularly at the east side of the orbit swath, are corrected by more than 0.3 for the new version 2.1 processor.

Figure 13 compares the ROCINN_CRB cloud pressures from the version 1.x and 2.1 processors for orbit 6939 on 14

February 2019. The cloud pressure differences are generally small for optically thin clouds with small cloud fractions. Due to the enlarged OCRA cloud fractions for relatively thick clouds in Fig. 11, the new ROCINN shows increased cloud pressures for large cloud fractions. The decreased cloud pressures e.g. over the Adriatic Sea is related to the reduction of surface albedo. Similar variations are observed for ROCINN_CAL cloud top and base pressures.

Figure 14 shows the tropospheric $NO_2$ columns for orbit 6939 on 14 February 2019 and the effect of upgrading the

OCRA/ROCINN processor from version 1.x to version 2.1. The tropospheric $NO_2$ columns reduce by more than $5 \times 10^{14}$ molec/cm$^2$ for the edge of the swath, such as the Po Valley, Rome, and Naples in Italy, and reduce by up to $3.5 \times 10^{14}$ molec/cm$^2$ for snow/ice scenarios, for instance the Ore Mountains. For polluted areas with optically thicker clouds (cloud frac-





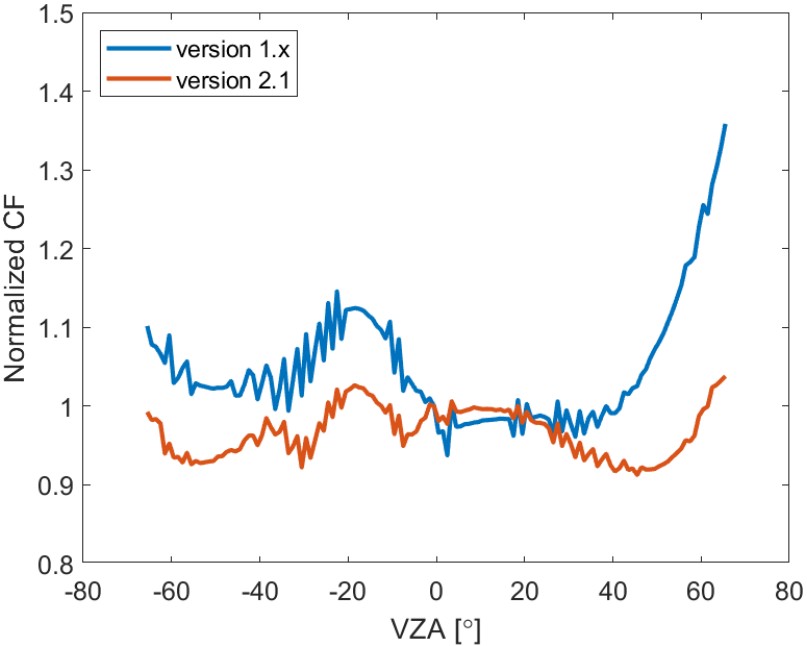

**Figure 12.** VZA dependency of OCRA cloud fractions from the version 1.x and 2.1 processors (normalized to nadir) in February 2019. Only measurements with $0 <$ cloud fraction $<= 0.4$ are included. The VZA is defined negative for observations on the west side of the orbit swath.

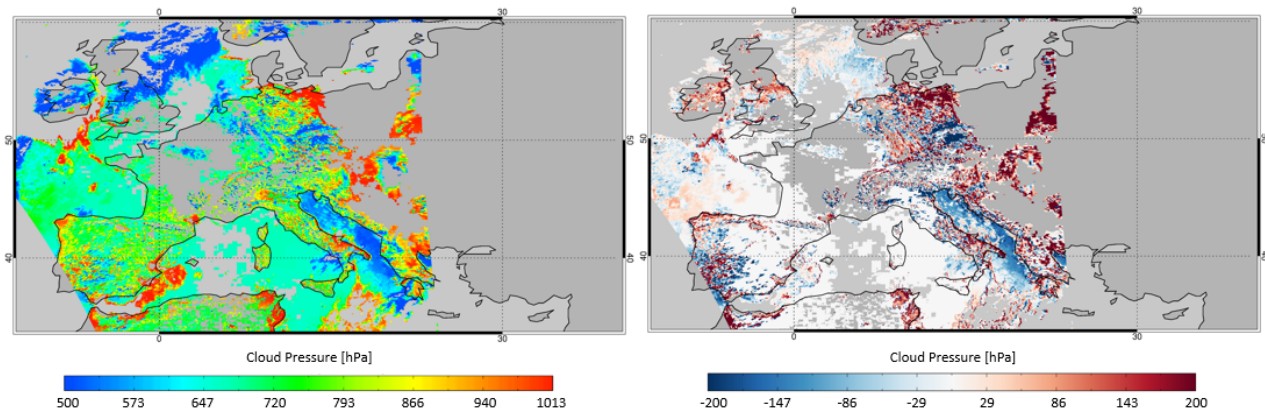

**Figure 13.** ROCINN_CRB cloud pressures from the version 2.1 processor (left) and comparisons with version 1.x (right) over Europe for orbit 6939 on 14 February 2019. Only measurements with $0 <$ cloud fraction $<= 0.4$ are included.

tion larger than 0.15 and cloud pressure larger than 700 hPa), e.g. northern Germany and the Benelux, the tropospheric $NO_2$ columns increase by more than $1 \times 10^{15}$ molec/cm$^2$, because the increase in cloud fraction (and thus cloud radiance fraction)

makes the retrieval less sensitive to the $NO_2$ below the cloud.



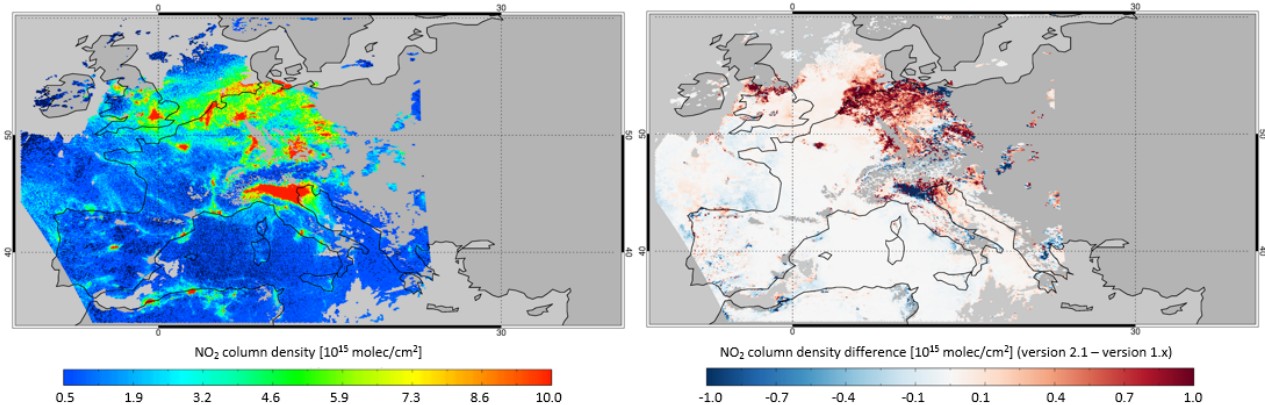

**Figure 14.** Tropospheric NO$_2$ columns retrieved using the OCRA/ROCINN cloud parameters from the version 2.1 processor (left) and differences with version 1.x (right) over Europe for orbit 6939 on 14 February 2019. Only measurements with a cloud radiance fraction less than 0.5 are included.

### 4.3.2 CAL cloud model

The cloud correction in our TROPOMI NO$_2$ retrieval is improved using the CAL model from the ROCINN cloud algorithm (Loyola et al., 2018). The CAL model, which regards the clouds as optically uniform layers of light-scattering water droplets, is more representative of the real situation than the CRB model, which treats the clouds as idealized Lambertian reflectors
with zero transmittance. The CAL model considers the multiple scattering of light inside the cloud and the contribution of the atmospheric layer between the cloud bottom and the ground.

Figure 15 presents the box-AMFs for clear and cloudy sky calculated using the CRB and CAL cloud models over Munich in Germany (48.09°N, 11.40°E) on 5 February 2019. The cloud pressures and the calculated tropospheric AMFs are also reported. Compared to the clear-sky box-AMFs, the cloudy-sky values increase above the cloud layer (albedo effect) and decrease below
the cloud layer (shielding effect). The CRB-based cloud retrieval generally shows a cloud height (pressure) close to the altitude of the middle (Ferlay et al., 2010; Richter et al., 2015), because CRB neglects the oxygen absorption within a cloud layer (Vasilkov et al., 2008) and misinterprets the smaller top-of-atmosphere reflectance as a lower cloud layer (Saiedy et al., 1967). Compared to the CRB-based cloud correction, the use of CAL model considers the sensitivities inside and below the cloud layers and increases the tropospheric AMFs by 0.09 (13.2%) for Munich.
Figure 16 presents the monthly average differences in the tropospheric NO$_2$ columns retrieved using the ROCINN_CRB and ROCINN_CAL cloud models in February and August 2019. The use of CAL cloud correction decreases the tropospheric NO$_2$ columns by more than $1 \times 10^{15}$ molec/cm$^2$ for polluted regions in winter, when most of the NO$_2$ concentrations are located at the surface (as shown in Fig. 9) and the cloud fractions are generally larger due to the seasonal variation of clouds. The effect is less than $5 \times 10^{14}$ molec/cm$^2$ for summer.





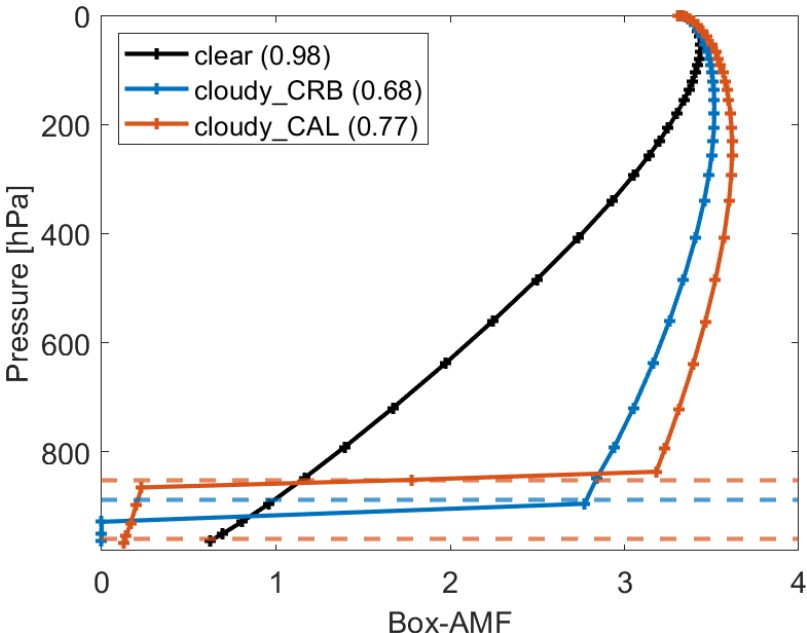

**Figure 15.** Box-AMFs for clear and cloudy sky using the ROCINN_CRB and ROCINN_CAL cloud models over Munich in Germany (48.09°N, 11.40°E) on 5 February 2019. The calculated tropospheric AMF is given in the bracket next to each label in the legend. The ROCINN_CRB cloud top pressure is shown as a blue horizontal dotted line, and the ROCINN_CAL cloud top and base pressures are shown as brown horizontal dotted lines.

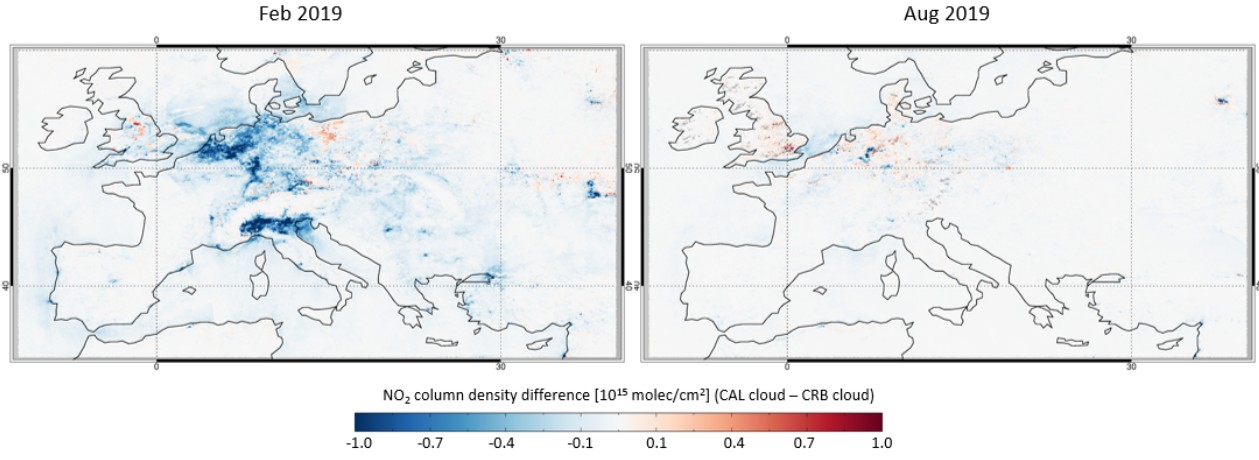

**Figure 16.** Differences in the tropospheric $NO_2$ columns retrieved using the ROCINN_CRB and ROCINN_CAL cloud models over Europe in February and August 2019. Only measurements with cloud radiance fraction less than 0.5 are included.



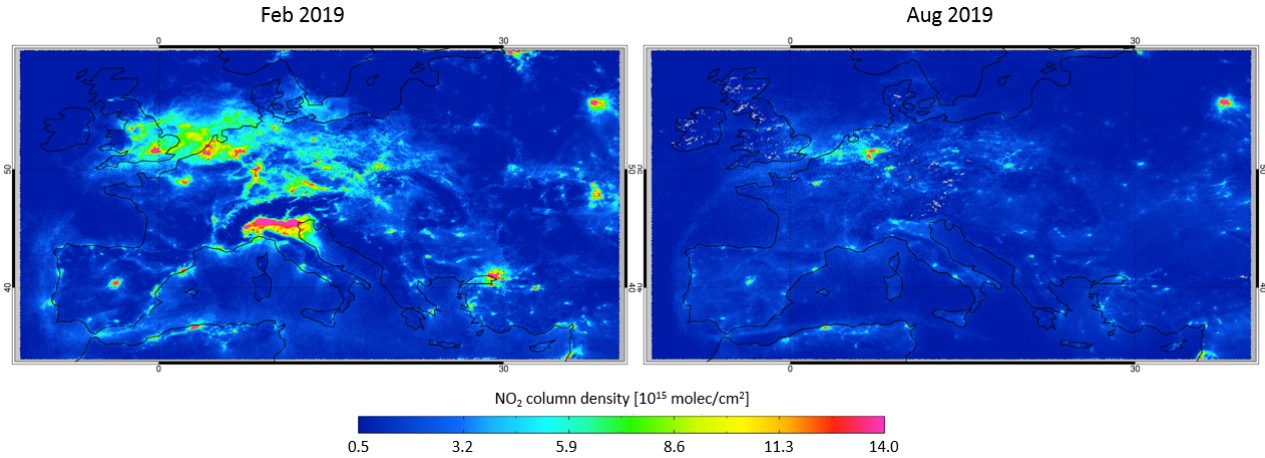

**Figure 17.** Tropospheric NO$_2$ columns from the improved algorithm over Europe in February and August 2019. Only measurements with cloud radiance fraction less than 0.5 are included.

## 5   TROPOMI tropospheric NO$_2$ measurements

### 5.1   Examples of TROPOMI tropospheric NO$_2$ measurements

Figure 17 shows the tropospheric NO$_2$ columns over Europe retrieved using the improved algorithm in February and August 2019. The tropospheric NO$_2$ columns are higher than $5 \times 10^{15}$ molec/cm$^2$ over urban and industrial areas in winter, such as the Po Valley, Germany's Ruhr region, the Benelux, South-East England, and Turkey's Marmara region. City-size polluted regions, e.g. around Paris, Madrid, Rome, Athens, and Moscow, are captured by the TROPOMI NO$_2$ measurements. NO$_2$ emissions over the shipping routes, e.g. the maritime connection between the Iberian Peninsula and North Africa, as well as emissions over the highways, e.g. the main East-West thoroughfare in Austria, are also detected.

Figure 18 compares the tropospheric NO$_2$ columns retrieved using the reference algorithm and the improved algorithm over Europe in February and August 2019. The tropospheric NO$_2$ columns are on average enhanced by $2 \times 10^{15}$ molec/cm$^2$ in winter and $8 \times 10^{14}$ molec/cm$^2$ in summer mainly due to the combined effect of the improvements in the AMF calculation. Larger differences by more than $3 \times 10^{15}$ molec/cm$^2$ are noticed in polluted regions, such as London, Paris, and the Po Valley, as well as shipping lanes, e.g. in the Mediterranean Sea.

### 5.2   Uncertainty estimates

Derived by uncertainty propagation (Boersma et al., 2004), the overall uncertainty on the tropospheric NO$_2$ column is directly related to the main retrieval steps, which are performed independently and assumed to be uncorrelated. The slant column uncertainty, estimated following a statistical method (Boersma et al., 2007) based on the spatial variability in the slant columns over the Pacific Ocean (20°S-20°N, 160°E-180°E), is on average $4.5 \times 10^{14}$ molec/cm$^2$. The uncertainty in the stratospheric

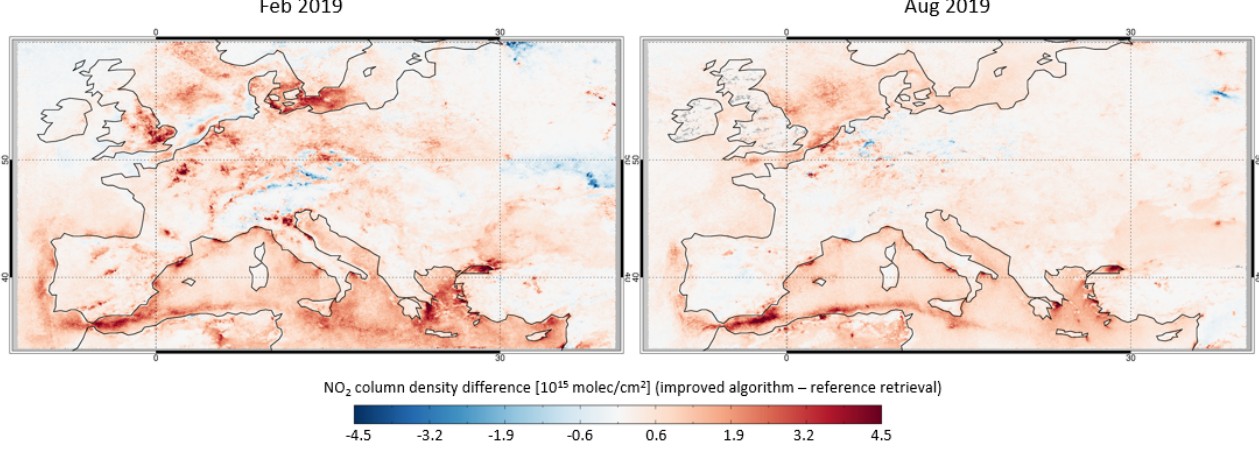

**Figure 18.** Differences in the tropospheric $NO_2$ columns retrieved using the reference algorithm and improved algorithm over Europe in February and August 2019. Only measurements with cloud radiance fraction less than 0.5 are included.

columns is $3.5 \times 10^{14}$ molec/cm$^2$ for polluted conditions based on the daily synthetic data (see Sect. 3.2) and $1 \times 10^{14}$ molec/cm$^2$ for monthly averages.

The tropospheric AMF calculation, which is the largest source of $NO_2$ uncertainty for polluted scenarios (Lorente et al., 2017), is mainly dependent on surface albedo, a priori $NO_2$ profile, cloud fraction, and cloud pressure, as introduced in Sect. 2.3 and 4. The tropospheric AMF uncertainties are calculated based on uncertainty propagation (Boersma et al., 2004) and typical uncertainties of each parameter (De Smedt et al., 2018, Table 8 therein).

Figure 19 shows the estimated tropospheric AMF uncertainties due to the errors in the surface albedo, cloud pressure, and
a prior $NO_2$ profile. The uncertainty contribution from the a prior $NO_2$ profile is practically described by a parameter referred to as profile height, defined as the altitude (pressure) below which resides 75% of the integrated $NO_2$ profile (De Smedt et al., 2018). As the satellite measurements are normally filtered for cloud radiance fraction smaller than 0.5 or cloud fraction smaller than ∼0.2, the uncertainties related to the cloud fraction are generally smaller than 15% (not shown). From Fig. 19, larger uncertainties are found for small albedo values and for scenarios with large albedo biases such as new snow/ice coverage. The
uncertainties due to the cloud pressure and a priori $NO_2$ profile can be up to 70% when the cloud is located below or within the $NO_2$ layer, particularly for thick clouds at low altitudes and for polluted situations (large profile heights).

The presence of aerosols can affect the sensitivity to tropospheric $NO_2$, depending on the particle properties and the $NO_2$ and aerosol vertical distribution (Martin et al., 2003; Leitão et al., 2010). The aerosol effect is not explicitly corrected in this study assuming that the effective cloud parameters from OCRA/ROCINN have partly accounted for the effect of aerosols on the light
paths (Boersma et al., 2004, 2011). In comparison to the simple CRB-based cloud correction, which can not fully describe the effects inherent to aerosol particles (Chimot et al., 2019), the use of CAL cloud correction considers the sensitivities inside and below the cloud/aerosol layers and reduces the AMF errors by more than 10% (Liu et al., 2020c).



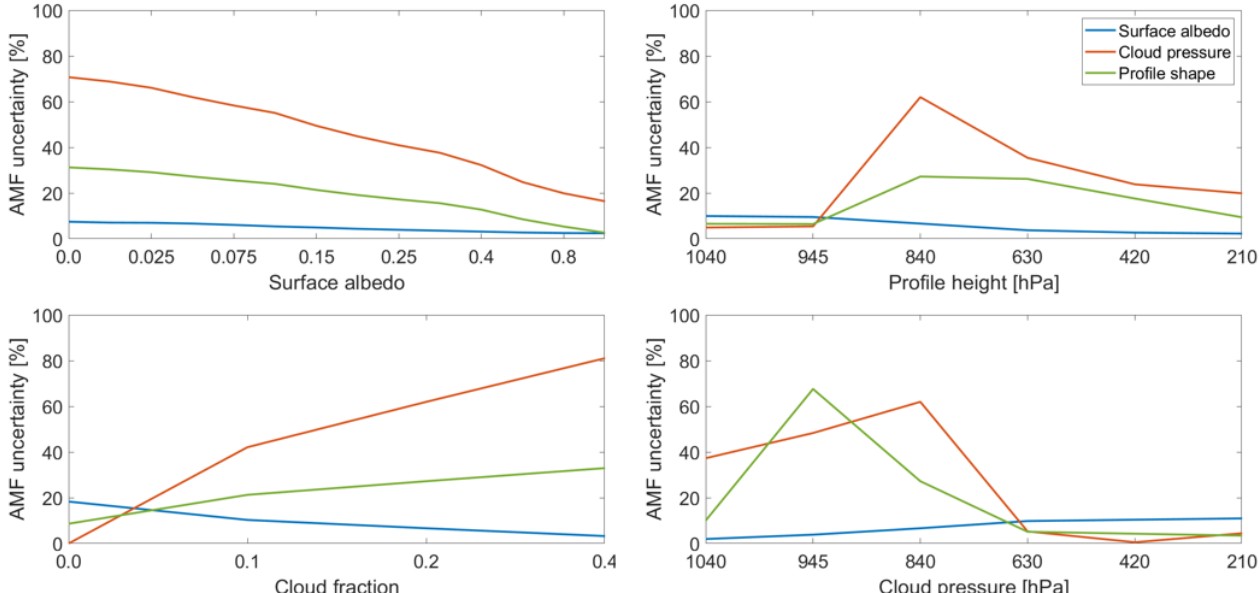

**Figure 19.** Tropospheric AMF uncertainties related to the surface albedo, cloud pressure, and a priori $NO_2$ profile errors. By default, the surface pressure is 1050 hPa, the surface albedo is 0.05, the profile height is 840 hPa, the cloud pressure is 840 hPa, the cloud fraction is 0.2. The definition of profile height is given in the text.

Note that the use of averaging kernel, which describes the vertical sensitivity of measurements of $NO_2$ concentrations, can remove the uncertainty contributed by the a priori $NO_2$ profile for applications such as data assimilation and validation study

(Eskes and Boersma, 2003). Therefore, for a typical polluted scene, the tropospheric AMF uncertainty is estimated to be 20% for mostly clear sky and 50% in the presence of clouds, leading to a total uncertainty in the tropospheric $NO_2$ columns in the 30-60% range.

## 6 TROPOMI tropospheric $NO_2$ validation

The validation of the improved TROPOMI tropospheric $NO_2$ columns is based on ground-based MAX-DOAS measurements

from nine stations in Europe. Table 3 provides the information about the stations, most of which are characterised by urban or suburban polluted conditions with heavy traffic and industrial emissions. For the validation of TROPOMI measurements, the satellite data from January 2018 - June 2020 are filtered for clouds (cloud radiance fraction less than 0.5), and the closest valid pixel within 20 km of the stations is compared to the ground-based MAX-DOAS data, which are linearly interpolated to the TROPOMI overpass time if original data exist within 1 h.

Figure 20 shows the time series and scatter plot of the comparison of the daily and monthly means between the improved TROPOMI tropospheric $NO_2$ columns and the ground-based MAX-DOAS measurements in Uccle. The monthly mean values from the TROPOMI and MAX-DOAS measurements show similar seasonal variations in the tropospheric $NO_2$ column. Figure





**Table 3.** An overview of MAX-DOAS stations contributing to the TROPOMI tropospheric NO$_2$ validation in this study. More details on the QA4ECV datasets can be found at http://www.qa4ecv.eu/ecvs.

| Station | Location | Institute | Description |
|---|---|---|---|
| Athen | 38.05°N, 23.86°E | IUPB | QA4ECV dataset |
| Bremen | 53.10°N, 8.85°E | IUPB | QA4ECV dataset |
| Cabauw | 51.97°N, 4.93°E | KNMI | Vlemmix et al. (2010) |
| De Bilt | 52.10°N, 5.18°E | KNMI | Vlemmix et al. (2010) |
| Mainz | 49.99°N, 8.23°E | MPIC | QA4ECV dataset |
| Munich | 48.15°N, 11.57°E | LMU | Chan et al. (2020) |
| Thessaloniki_ciri | 40.56°N, 22.99°E | AUTH | Drosoglou et al. (2017), QA4ECV dataset |
| Thessaloniki_lap | 40.63°N, 22.96°E | AUTH | Drosoglou et al. (2017), QA4ECV dataset |
| Uccle | 50.80°N, 4.36°E | BIRA-IASB | Gielen et al. (2014), Hendrick et al. (2014), Dimitropoulou et al. (2020) |

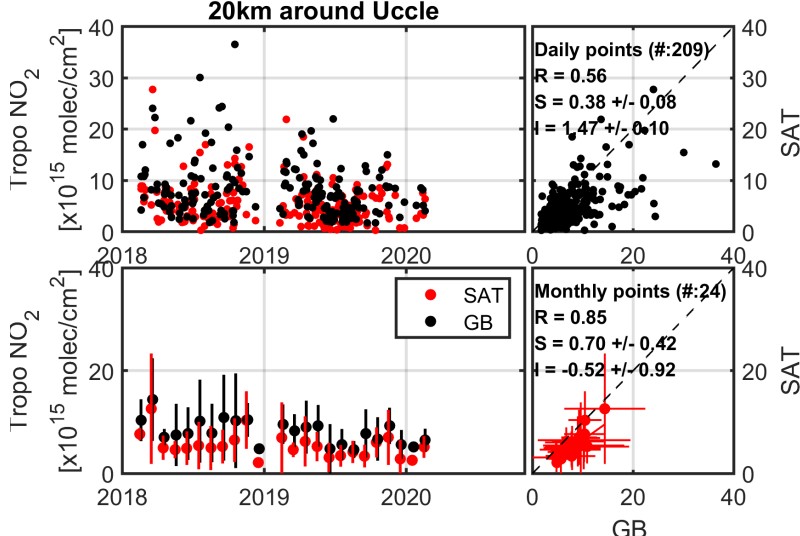

**Figure 20.** Daily and monthly mean time series and scatter plot of TROPOMI (SAT) and MAX-DOAS (GB) tropospheric NO$_2$ columns (closest valid pixel within 20 km of Uccle). Results are shown for the improved satellite retrieval algorithm.

20 includes the statistical information on the Pearson correlation coefficient as well as the slope and intercept obtained with the robust Theil–Sen estimator (Sen, 1968; Vigouroux et al., 2020). A correlation coefficient of 0.85, a slope of 0.70, and an intercept of $-0.52 \times 10^{15}$ molec/cm$^2$ are derived when comparing the monthly mean values.

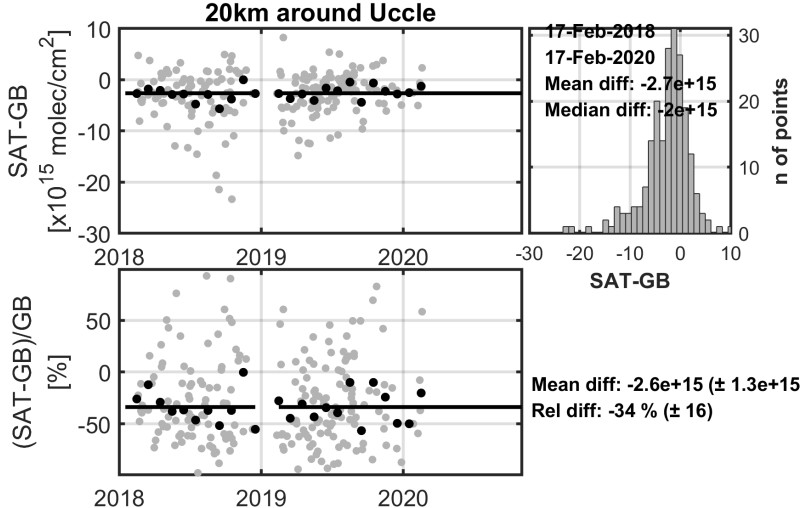

**Figure 21.** Daily (grey dots) and monthly mean (black dots) absolute and relative TROPOMI (SAT) and MAX-DOAS (GB) time series differences for the Uccle station. Results are shown for the improved satellite retrieval algorithm. The histogram of the daily differences is also given, showing the mean and median difference. The total mean values of absolute and relative monthly differences are given in the bottom-right panel.

Figure 21 presents the daily and monthly mean absolute and relative differences of TROPOMI and MAX-DOAS measurements in Uccle. The differences are generally within $1 \times 10^{16}$ molec/cm² with a mean difference of $-2.6 \times 10^{15}$ molec/cm². The NO₂ levels are underestimated by 34% by TROPOMI with a standard deviation of 16%, which is mostly explained by the relatively low sensitivity of spaceborne measurements near the surface, the aerosol shielding effect, and the gradient smoothing
effect. These effects are often inherent to the remaining impact of structural uncertainties (Boersma et al., 2016), such as the impact of the choice of the a priori NO₂ profiles and/or the albedo database assumed for the satellite AMF calculations, and to the different measurement types or the specific conditions of the validation sites.

To analyse the gradient smoothing effect for Uccle, TROPOMI measurements for 2018-2020 are aggregated based on an area-weighted tessellation to a resolution of $0.01° \times 0.01°$, and the systematic variation in tropospheric NO₂ columns between
the satellite pixel location and the ground-based station position is shown in Fig. 22, following the method from Chen et al. (2009); Ma et al. (2013); Pinardi et al. (2020). From Fig. 22, the smoothing effect is largest for summer (up to 19%), as the NO₂ gradients are large due to the shorter lifetime, in agreement with Ma et al. (2013). For the Uccle site, which is located south of Brussels at a distance of ∼6 km from the city center, the tropospheric NO₂ columns increase by up to 4% outwards until 6 km due to the influence of the surrounding emission sources during summer and autumn. This effect is additionally
influenced by the seasonal wind pattern, particularly for winter, when the wind is blowing in the direction of the site from north (Dimitropoulou et al., 2020).





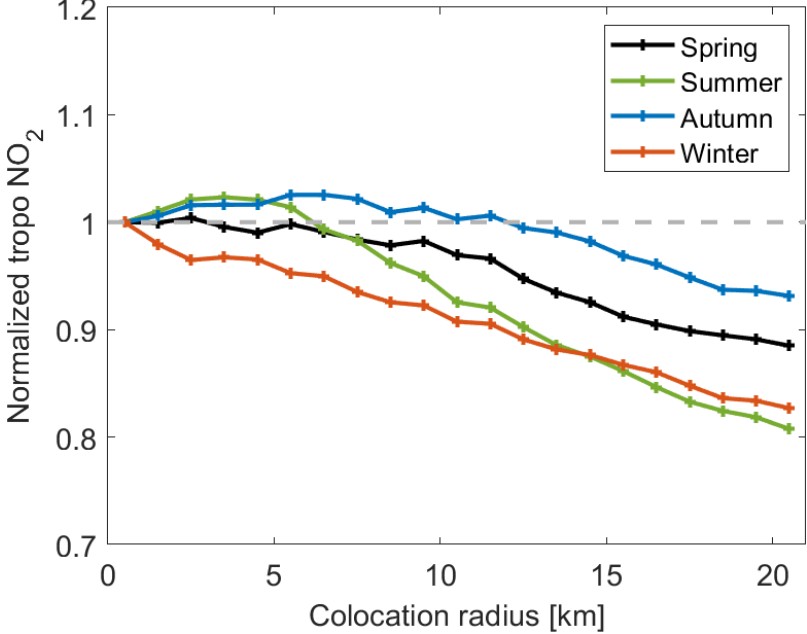

**Figure 22.** Normalized tropospheric NO$_2$ columns at increasing colocation radii for the Uccle station, estimated using seasonal mean TROPOMI data in 2018-2020.

**Table 4.** The mean difference (MD, SAT-GB in $1 \times 10^{15}$ molec/cm$^2$), standard deviation (STD, in $1 \times 10^{15}$ molec/cm$^2$), relative difference (RD, in %), the Pearson correlation coefficient R, as well as the slope S and intercept I (in $1 \times 10^{15}$ molec/cm$^2$) obtained with the robust Theil–Sen estimator for the monthly TROPOMI tropospheric NO$_2$ product compared to MAX-DOAS data. Stations are ordered by increasing mean difference. Values for the improved algorithm are given and the values for the reference algorithm are reported in brackets for comparison.

| Station | MD | STD | RD | R | S | I |
|---|---|---|---|---|---|---|
| Athens | -1.6 (-3.5) | 1.2 (1.4) | -26 (-53) | 0.85 (0.87) | 0.65 (0.42) | 0.33 (0.25) |
| De Bilt | -2.0 (-3.5) | 1.8 (1.6) | -27 (-51) | 0.63 (0.72) | 0.42 (0.25) | 1.69 (1.31) |
| Thessaloniki_ciri | -2.2 (-3.4) | 1.5 (2.0) | -34 (-54) | 0.91 (0.95) | 0.51 (0.45) | -0.80 (0.14) |
| Thessaloniki_lap | -2.4 (-3.7) | 2.6 (2.6) | -27 (-49) | 0.79 (0.86) | 0.35 (0.30) | 1.97 (1.09) |
| Bremen | -2.6 (-3.7) | 1.0 (1.2) | -45 (-61) | 0.85 (0.71) | 0.58 (0.44) | -0.24 (-0.20) |
| Uccle | -2.6 (-4.5) | 1.3 (1.5) | -34 (-55) | 0.85 (0.81) | 0.70 (0.42) | -0.52 (-0.24) |
| Cabauw | -3.2 (-4.7) | 1.8 (2.0) | -40 (-59) | 0.75 (0.67) | 0.41 (0.21) | 1.08 (1.24) |
| Munich | -3.4 (-4.6) | 2.5 (2.3) | -39 (-56) | 0.57 (0.72) | 0.35 (0.39) | 1.44 (0.29) |
| Mainz | -4.4 (-5.6) | 3.0 (2.8) | -40 (-60) | 0.85 (0.87) | 0.39 (0.28) | 1.65 (0.82) |

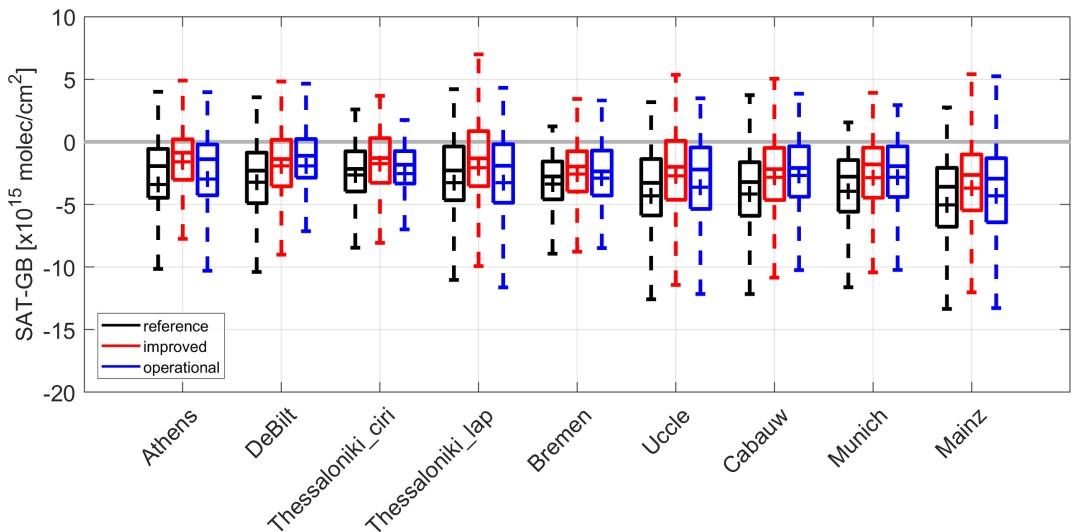

**Figure 23.** Box and whisker plot of the daily biases and spread of the differences between the TROPOMI (SAT) and MAX-DOAS (GB) data. Results for the reference algorithm, the improved algorithm, and the operational product are compared. Stations are ordered by increasing mean difference. The mean differences are represented by crosses. The median differences are represented by vertical solid lines inside the boxes, which mark the 25 and 75% quantiles. The whiskers cover the 9-91% range of the differences.

Similar figures as Fig. 20 and 21 for the improved and reference algorithms are gathered in Fig. S1 - S4 in the Supplement for all the stations. Figure 23 shows an overview of the daily differences between satellite and ground-based data for the improved and reference algorithms. Table 4 summarizes the monthly comparisons of TROPOMI and MAX-DOAS measurements. High
correlations are observed for the improved algorithm for all the stations with an average correlation coefficient of 0.78. The impact of the algorithm improvements leads to a decrease of the mean absolute difference in urban/suburban conditions from $-4.13 \times 10^{15}$ molec/cm$^2$ to $-2.71 \times 10^{15}$ molec/cm$^2$ and relative difference from -55.3% to -34.7%. The largest absolute bias ($-5.6 \times 10^{15}$ molec/cm$^2$ in Mainz) is reduced to $-4.4 \times 10^{15}$ molec/cm$^2$ (relative bias from -60% to -40%), while the smaller absolute bias ($-3.4 \times 10^{15}$ molec/cm$^2$ in Thessaloniki_ciri) is reduced to $-2.2 \times 10^{15}$ molec/cm$^2$ (relative bias from -54% to
-34%). The largest reduction is found for Athens (-27% reduction from the reference to improved algorithm).

Smaller biases are found for the improved algorithm, not only in comparison with the reference algorithm but also compared to the operational product in Fig. 23, particularly for Athens, Thessaloniki_ciri, Thessaloniki_lap, Uccle, and Mainz. The relative biases ranging from -26 to -45% in Table 4 are lower than those reported by validation exercises for the operational TROPOMI product, where the NO$_2$ levels are normally found to be underestimated by the TROPOMI instrument by 30%
to 50% for polluted conditions (Dimitropoulou et al., 2020; Verhoelst et al., 2020; Wang et al., 2020). These results are not directly comparable to results e.g. obtained by Dimitropoulou et al. (2020), as they use a more elaborated ground-based dataset with several pointing directions and specific area-weighted pixel selections in the MAX-DOAS line-of-sight. Note that the



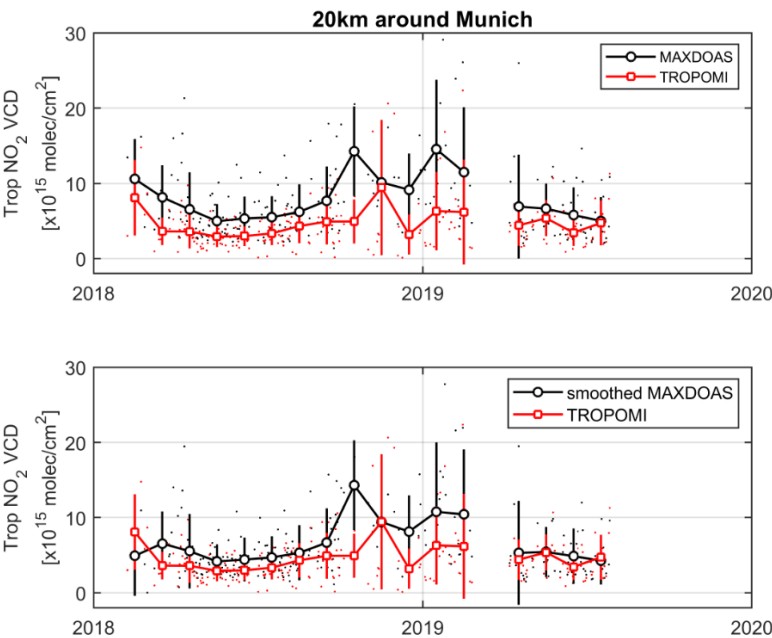

**Figure 24.** Daily (dots) and monthly mean (circles) time series of TROPOMI and MAX-DOAS tropospheric NO₂ columns for the Munich station. Results for the original comparisons and the smoothed comparisons are reported.

operational tropospheric NO$_2$ columns have been increased with an upgrade of the NO$_2$ processor (version 01.04.00) since 29 November 2020 due to the improved handling of cloud pressures (Eskes et al., 2020).

To investigate the impact of satellite a priori NO$_2$ profiles on the comparison, the satellite averaging kernel (see Sect. 5.2) is used to relate the MAX-DOAS retrieved NO$_2$ profiles to satellite column measurements by calculating the smoothed MAX-DOAS columns as:

$$V_{GB,smoothed} = \sum_l AK_{SAT,l} \times x_{GB,l}. \tag{8}$$

The smoothed MAX-DOAS NO$_2$ columns $V_{GB,smoothed}$ are derived by convolving the layer ($l$)-dependent daily profile $x_{GB,l}$
(expressed in partial columns and interpolated to the satellite overpass time) with the satellite averaging kernel $AK_{SAT,l}$.

    Figure 24 shows the original and smoothed comparisons of satellite and MAX-DOAS data for the Munich station. The use of the averaging kernel smoothing reduces the MAX-DOAS columns and thus improves the agreement between the satellite and MAX-DOAS columns. When the satellite averaging kernels are used to remove the contribution of the a priori NO$_2$ profile shape, the mean absolute difference reduces from $-3.4 \times 10^{15}$ molec/cm$^2$ to $-1.9 \times 10^{15}$ molec/cm$^2$, and the relative difference
reduces from -39% to -23%.





## 7 Conclusions

The DLR retrieval algorithm developed for TROPOMI $NO_2$ measurements follows a three-step scheme. To calculate the $NO_2$ slant columns, a 405-465 nm fitting window is applied in the DOAS fit for consistency with other $NO_2$ retrievals from OMI and TROPOMI. Absorption cross-sections of interfering species and a linear intensity offset correction are applied. The striping pattern of slant columns is corrected using an empirical method based on the daily averaged across-track variability of $NO_2$ slant columns over clean regions.

The stratospheric $NO_2$ component is estimated using the STREAM method, which requires no chemistry transport model data as used in data assimilation and provides an improved treatment of polluted and cloudy pixels comparing to other modified reference sector methods. An improved DSTREAM method is used to correct for the VZA dependency of stratospheric $NO_2$ for high latitudes, which is related to the local time changes across the orbit. DSTREAM divides the orbit swath into three segments, applies the original STREAM to data from each of the segments, and calculates the stratospheric $NO_2$ column based on VZA and latitude from the satellite measurement. Applied to synthetic TROPOMI data, constructed using the IFS(CB05BASCOE) model fields, the estimated stratospheric $NO_2$ columns from the original STREAM and the improved DSTREAM show good consistency with the a priori truth. Applied to actual TROPOMI measurements, STREAM and DSTREAM successfully separate the stratospheric and tropospheric fields for polluted regions. The VZA dependency of stratospheric $NO_2$ which amounts up to $2 \times 10^{14}$ molec/cm$^2$ at high latitudes is captured by DSTREAM.

In the tropospheric AMF calculation, the surface albedo from the monthly OMI LER climatology is replaced by the TROPOMI GE_LER data, which is consistently applied in both $NO_2$ and cloud retrievals. GE_LER in the $NO_2$ fitting window is retrieved using the machine learning based approach FP_ILM with inputs from the DOAS fitting. In comparison with the climatological LER values from previous satellite missions, the GE_LER data relies on the real-time measurements from the TROPOMI instrument itself with an improved spatial resolution of $0.1° \times 0.1°$. Therefore, GE_LER better characterizes the actual surface conditions with an impact on the tropospheric $NO_2$ columns by up to $3 \times 10^{15}$ molec/cm$^2$ under polluted conditions.

Mesoscale-resolution a priori profiles ($0.2° \times 0.3°$), obtained from the regional POLYPHEMUS/DLR chemistry transport model based on the European TNO-MACC_II emission inventory, provide a better description of the spatial variability in the $NO_2$ fields for Europe. Compared to the currently used TM5-MP profiles, the POLYPHEMUS/DLR profiles generally show higher surface $NO_2$ concentrations, which reduce the tropospheric AMFs and thus enhance the tropospheric $NO_2$ columns by more than $2 \times 10^{15}$ molec/cm$^2$ for polluted regions.

The presence of clouds is considered using the TROPOMI operational cloud retrieval algorithms OCRA/ROCINN. In a new version 2.1 processor, OCRA separates a spectral scene (in the UV-VIS wavelength range) into cloudy contribution and cloud-free background using TROPOMI-based background maps ($0.1° \times 0.1°$) instead of OMI-based ones, and ROCINN applies the surface albedo from the GE_LER data in the TROPOMI NIR instead of a static climatology. The overestimation of cloud fractions at the swath edge is corrected. Larger differences in cloud fractions and cloud pressures are found for relatively thick clouds, which affect the tropospheric $NO_2$ columns by more than $1 \times 10^{15}$ molec/cm$^2$. In the tropospheric AMF calculation,





the CRB model from ROCINN, in which clouds are idealized Lambertian reflectors, is replaced with the CAL model, in which clouds are represented by uniform layers of water droplets. CAL is more representative of the real situation and preferred for small TROPOMI ground pixels and for low clouds. The application of CAL cloud parameters considers the sensitivities inside and below the cloud layers and reduces the tropospheric $NO_2$ columns by more than $1 \times 10^{15}$ molec/cm$^2$ for polluted regions.

    The uncertainty in the $NO_2$ slant columns is $4.5 \times 10^{14}$ molec/cm$^2$, derived from the spatial variability over the Paficic Ocean.

The uncertainty in the stratospheric columns is $3.5 \times 10^{14}$ molec/cm$^2$ for polluted regions based on daily synthetic TROPOMI data. The tropospheric AMF uncertainty is estimated to be 20% for mostly clear sky and 50% in the presence of clouds, leading to a total uncertainty in the tropospheric $NO_2$ column in the 30-60% range.

    Validation of the improved TROPOMI tropospheric $NO_2$ columns is performed by comparisons with ground-based MAX-DOAS measurements. The validation is illustrated for nine European stations with urban/suburban conditions. The improved

data shows a similar seasonal variation in the tropospheric $NO_2$ columns as the MAX-DOAS measurements with an average correlation coefficient of 0.78. Compared to the reference data, the improved algorithm shows a significant improvement with absolute differences decreasing from $-4.13 \times 10^{15}$ molec/cm$^2$ to $-2.71 \times 10^{15}$ molec/cm$^2$ on average and relative differences from -55.3% to -34.7%. When the satellite averaging kernels are used to remove the contribution of a priori $NO_2$ profile shape, the absolute difference at the Munich station reduces from $-3.4 \times 10^{15}$ molec/cm$^2$ to $-1.9 \times 10^{15}$ molec/cm$^2$, and the relative

difference reduces from -39% to -23%.

    In the future, the spectral effect of extending the fitting window to 490 nm will be analysed, when the pixel blooming is better treated in a future update of the TROPOMI level 0-1b processor. The operational OCRA/ROCINN cloud parameters will be compared with other cloud products such as FRESCO-S and MICRU. The interpretation of the cloud product for aerosol-dominated scenes and the impact on the $NO_2$ retrieval algorithm will be further investigated. Aerosol contamination

will be removed in the GE_LER retrieval using TROPOMI aerosol index data. The $NO_2$ data quality will be further analysed using data from additional ground-stations covering different pollution conditions and data from validation campaigns with independent instruments.

    The $NO_2$ retrieval algorithm can be adapted for new instruments and missions, such as the polar-orbiting Sentinel-5 and geostationary Sentinel-4 missions, which offer new perspectives for monitoring $NO_2$ with a fast revisiting time and a high

spatial resolution and provide information on atmospheric variables in support of European policies.

*Data availability.* The TROPOMI $NO_2$ datasets used in the study are available upon request.

*Author contributions.* SL and PV developed the retrieval framework. SL processed the data, analysed the results, and contributed to the GE_LER processing and MAX-DOAS validation. SL and GP analysed the MAX-DOAS validation results. JX processed the GE_LER data. AA and RL provided expertise regarding the OCRA/ROCINN-based cloud correction. KLC contributed to uncertainty analysis. SB

contributes to the STREAM development and improvement. EK and FB provided the POLYPHEMUS/DLR model data. VH provided the



IFS(CB05BASCOE) model data. AB, KLC, SD, SD, MG, FH, DK, KL, AP, JR, AR, MVR, TW, and MW provided the MAX-DOAS measurements. DL is the main developer of the TROPOMI cloud products based on the CAL model and the GE_LER retrievals used in this study. SL, PV, JX, and GP wrote the paper with comments from all authors.

*Competing interests.* The authors declare that they have no conflict of interest.

*Acknowledgements.* This work is funded by the DLR/DAAD Research Fellowships - Postdocs programme (57478192). The TROPOMI NO$_2$ data generated at DLR are used in the S-VELD project, which is financed by the Federal Ministry of Transport and Digital Infrastructure to analyse the effect of traffic emission on air quality in Germany (grant no. 19F2065). Part of the results discussed in this paper were achieved within the JOSEFINA project funded by the Bavarian State Ministry for Environment and Consumer Protection. The POLYPHMEMUS/DLR model was developed with support by the 7$^{th}$ EU Framework Program (PASODOBLE project grant 241557). We acknowledge the Belgian
Science Policy Office (BELSPO) for supporting part of this work through the PRODEX programme B-ACSAF project. Part of this work is carried out in the framework of the S5p Validation Team (S5PVT) AO projects NIDFORVAL (ID #28607, PI G. Pinardi, BIRA-IASB). Part of this work is supported by the DFG Major Research Instrumentation Programme (grant no. INST 86/1499-1 FUGG). We are grateful to QA4ECV for the generation of harmonized MAX-DOAS datasets. We thank EU/ESA/KNMI/DLR for the provision of the TROPOMI/S5P level 1 products. We thank DLR colleagues for developing the Universal Processor for UV/Vis Atmospheric Spectrometers (UPAS) system
used for generating level 2 products from TROPOMI. This paper contains modified Copernicus Sentinel data processed by DLR.



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
