# Peer review of "Figure S1. Daily and monthly mean time series and scatter plot of TROPOMI and MAXDOAS tropospheric NO2 columns (closest valid pixel within 20 km) over Athens, Bremen, Cabauw, De Bilt, Mainz, Munich, Thessaloniki\_ciri, and Thessaloniki\_lap. Results are shown for the improved satellite re"

_Atmospheric Measurement Techniques, 2021_

## Author Comment (AC1)

**Response to Referee Henk Eskes**

We thank the reviewer Henk Eskes for the careful reading and the usefulmments. Below we give the reviewer's comment, our response, and the changed text in the manuscript. The page and line numbers refer to the mark-up version of the manuscript.

**General remarks:**

**1.The reader may find the title and abstract confusing. It suggests that the authors are improving the operational TROPOMI retrieval product, while actually the paper discusses and upgrade of the retrieval product implemented at DLR. This should be made more clear at the top of the abstract and in the title. I suggest the authors make clear which retrieval product they refer to, for instance by calling it the DLR TROPOMI NO₂ European product. Maybe good to add the word "scientific" as opposed to the "operational" retrieval.**

We have updated the title to "*An improved TROPOMI tropospheric NO₂ research product over Europe*" and referred the data to "*the DLR TROPOMI NO₂ European product*" throughout the paper.

**2.The preformance of the POLYPHEMUS model is not easy to judge. The resolution of this model is not very high, 0.2x0.3 degree, while TROPOMI has a resolution of 0.05 degree. The CAMS models for instance run at 0.1x0.1 degree. This resolution is in between the TM5-MP and TROPOMI. Would higher resolution produce a major further improvement? I would like to see a comparison between POLYPHEMUS and TROPOMI as extra figure. The POLYPHEMUS model seems to produce very low free tropospheric NO₂ concentrations. Is this expected to impact the retrieval?**

The 0.2x0.3 degree is selected for POLYPHEMUS/DLR due to the feasibility in terms of runtime and affordability in terms of computing resources. We have analysed the effect of model resolution using a priori NO₂ profiles from LOTOS-EUROS, which constitutes one of the state-of-the-art atmospheric chemistry models used by the regional Copernicus Atmosphere Monitoring Service (CAMS, http://www.copernicus-atmosphere.eu). Figure R1 shows the differences in the tropospheric NO₂ columns retrieved by altering the model resolution for LOTOS-EUROS (0.1x0.1 degree - 0.2x0.3 degree) in February 2019. The increase of the spatial resolution from 0.2x0.3 to 0.1x0.1 degree improves the tropospheric NO₂ columns moderately by up to $5\times10^{14}$ molec/cm$^2$ or 11% for polluted regions.

[Figure]

Figure R1. Differences in the tropospheric NO$_2$ columns retrieved by altering the model resolutions (0.1x0.1 degree - 0.2x0.3 degree) for LOTOS-EUROS a priori NO$_2$ profiles in February 2019.

We have included a new Fig. 11 in page 20 comparing the tropospheric NO$_2$ columns from TROPOMI measurements and POLYPHEMUS/DLR simulations (using the satellite averaging kernel) and included the description in pages 19 as "*Figure 11 compares the tropospheric NO$_2$ columns from TROPOMI measurements and POLYPHEMUS/DLR simulations. The satellite averaging kernel, which describes the vertical sensitivity of measurements of NO$_2$ concentrations, is applied to reduce the systematic biases caused by unrealistic a priori profile information. From Fig. 11, POLYPHEMUS/DLR NO$_2$ columns are generally higher than satellite measurements, which can be partly related to the use of the TNO-MACC emission dataset (Denier van der Gon et al., 2010). An update of the POLYPHEMUS/DLR model using the more recent TNO-MACC_II emission (Kuenen et al., 2014) is planned for the near future. We note here that the profile shape is of far more importance than the column bias for the interpretation of satellite retrievals.*"

[Figure]

Figure 11. Tropospheric NO₂ columns from TROPOMI measurements (retrieved with the POLYPHEMUS/DLR a priori NO₂ profiles) and POLYPHEMUS/DLR simulations (using the satellite averaging kernel) over Europe in February and August 2019. Only TROPOMI measurements with cloud radiance fraction less than 0.5 are included.

Despite the use of outdated emissions, the POLYPHEMUS/DLR is in general reliable, as indicated in a validation exercise in Fig. R2 comparing the near-surface NO₂ concentrations from POLYPHEMUS/DLR and ground-based EEA air quality monitoring stations (https://discomap.eea.europa.eu/map/fme/AirQualityExport.htm). The correlation coefficients are generally higher than 0.6 for polluted hot spots, and the biases are generally lower than 20 uq/m³ or 38%.

[Figure]

Figure R2. Comparisons of near-surface NO$_2$ concentrations [uq/m$^3$] from POLYPHEMUS/DLR simulations and ground-based air quality monitoring stations (POLYPHEMUS/DLR results – ground-based observations) for Europe in 2018.

[Figure]

Figure R3. A priori $NO_2$ profiles from the chemistry transport models LOTOS-EUROS, POLYPHEMUS/DLR, and TM5-MP over Hamburg in Germany (53.55°N, 9.99°E) on 1 February 2019. The calculated clear-sky tropospheric AMF is given in the bracket next to each label in the legend. Normalized profiles (to the lowest values) are also shown on a logarithm scale.

We have compared a priori $NO_2$ profiles from LOTOS-EUROS, POLYPHEMUS/DLR, and TM5-MP over Hamburg in Fig. R3. POLYPHEMUS/DLR shows the largest surface layer $NO_2$ concentration (Fig. R3 left panel) and the steepest profile shape (Fig. R3 right panel), which yields the smallest tropospheric AMF. LOTOS-EUROS shows lower free tropospheric concentration and larger surface concentration than TM5-MP, and thus the tropospheric AMF is lower by 9%.

[Figure]

Figure R4. Differences in the tropospheric $NO_2$ columns retrieved using the POLYPHEMUS/DLR and LOTOS-EUROS a priori $NO_2$ profiles over Europe in February and August 2019.

Figure R4 compares the tropospheric $NO_2$ columns retrieved using POLYPHEMUS/DLR and LOTOS-EUROS a priori $NO_2$ profiles, where the use of LOTOS-EUROS reduces the tropospheric $NO_2$ columns by up to $\sim 1 \times 10^{15}$ molec/cm$^2$, e.g. over the ocean partly due to difference in the shipping emissions.

**3.It would be of interest to extend the comparisons with the operational TROPOMI product and also list differences with other (regional) retrievals like the TROPOMI-POMINO approach for Asia. Maybe in the form of a table listing the choices for albedo, cloud, stratosphere, a-priori for several retrievals. For the validation, Table 4, it would be nice if also the operational retrieval could be added (now there is only the comparison with the old DLR retrieval in brackets).**

We have extended the comparison with different retrievals in Table 1 in page 9.

*Table 1. Overview of tropospheric $NO_2$ column retrievals. See Table 2 for details of the chemistry transport models used to obtain the a priori $NO_2$ profiles for the DLR improved retrieval in this work.*

|  | DLR reference retrieval | DLR improved algorithm | KNMI operational product | POMINO-TROPOMI product | ECCC product |
|---|---|---|---|---|---|
| reference | This work | This work | (van Geffen et al., 2020a, b) | (Lin et al., 2014; Liu et al., 2019a) | (Griffin et al., 2019) |
| Region | Global | Europe | Global | China | Canada |
| Stratospheric correction | STREAM | DSTREAM | Data assimilation | Same as KNMI | Same as KNMI |
| Surface albedo | OMI LER climatology | TROPOMI GE_LER data | OMI LER climatology | MODIS BRDF | MODIS albedo climatology |
| A priori $NO_2$ profile | TM5-MP | POLYPHEMUS/DLR | TM5-MP | GEOS-Chem | GEM-MACH |
| Cloud parameter | OCRA/ROCINN_CRB version 1.x | OCRA/ROCINN_CAL version 2.1 | FESCO-S | Recalculated cloud fraction with aerosol correction | Same as KNMI |
| Aerosol treatment | Implicit correction | Implicit correction | Implicit correction | Explicit correction | Explicit correction |

We have included the validation results for the operational product in Tables 5 and 6 in pages 31-32.

*Table 5. The Pearson correlation coefficient R, as well as the slope S and intercept I (in $1\times10^{15}$ molec/cm$^2$) obtained with the robust Theil–Sen estimator for the monthly TROPOMI tropospheric $NO_2$ product compared to MAX-DOAS data. Stations are ordered by increasing mean difference. Values for the DLR improved algorithm (DLRimp) are given, and the values for the DLR reference algorithm (DLRref) and the KNMI operational product (KNMIop) are reported for comparison.*

| Station | R | | | S | | | I | | |
|---|---|---|---|---|---|---|---|---|---|
| | DLRimp | DLRref | KNMIop | DLRimp | DLRref | KNMIop | DLRimp | DLRref | KNMIop |
| Athens | 0.85 | 0.87 | 0.81 | 0.65 | 0.42 | 0.45 | 0.33 | 0.25 | 0.33 |
| De Bilt | 0.63 | 0.72 | 0.80 | 0.42 | 0.25 | 0.77 | 1.69 | 1.31 | -0.57 |
| Thessaloniki_ciri | 0.91 | 0.95 | 0.91 | 0.51 | 0.45 | 0.26 | -0.80 | 0.14 | 1.23 |
| Thessaloniki_lap | 0.79 | 0.86 | 0.90 | 0.35 | 0.30 | 0.37 | 1.97 | 1.09 | 0.93 |
| Bremen | 0.85 | 0.71 | 0.70 | 0.58 | 0.44 | 0.40 | -0.24 | -0.20 | 0.79 |
| Uccle | 0.85 | 0.81 | 0.44 | 0.70 | 0.42 | 0.27 | -0.52 | -0.24 | 2.26 |
| Cabauw | 0.75 | 0.67 | 0.43 | 0.41 | 0.21 | 0.33 | 1.08 | 1.24 | 1.98 |
| Munich | 0.57 | 0.72 | 0.63 | 0.35 | 0.39 | 0.42 | 1.44 | 0.29 | 1.35 |
| Mainz | 0.85 | 0.87 | 0.81 | 0.39 | 0.28 | 0.50 | 1.65 | 0.82 | 0.02 |

*Table 6. Similar as Table 5 but for the mean difference (MD, SAT-GB in $1\times10^{15}$ molec/cm$^2$), standard deviation (STD, in $1\times10^{15}$ molec/cm$^2$), and relative difference (RD, in %) for the monthly TROPOMI tropospheric NO$_2$ product compared to MAX-DOAS data.*

| Station | MD | | | STD | | | RD | | |
|---|---|---|---|---|---|---|---|---|---|
| | DLRimp | DLRref | KNMIop | DLRimp | DLRref | KNMIop | DLRimp | DLRref | KNMIop |
| Athens | -1.6 | -3.5 | -3.1 | 1.2 | 1.4 | 1.3 | -26 | -53 | -48 |
| De Bilt | -2.0 | -3.5 | -2.1 | 1.8 | 1.6 | 1.3 | -27 | -51 | -30 |
| Thessaloniki_ciri | -2.2 | -3.4 | -3.8 | 1.5 | 2.0 | 4.0 | -34 | -54 | -51 |
| Thessaloniki_lap | -2.4 | -3.7 | -3.5 | 2.6 | 2.6 | 2.3 | -27 | -49 | -46 |
| Bremen | -2.6 | -3.7 | -3.1 | 1.0 | 1.2 | 1.4 | -45 | -61 | -46 |
| Uccle | -2.6 | -4.5 | -4.3 | 1.3 | 1.5 | 2.9 | -34 | -55 | -45 |
| Cabauw | -3.2 | -4.7 | -2.9 | 1.8 | 2.0 | 2.1 | -40 | -59 | -36 |
| Munich | -3.4 | -4.6 | -3.3 | 2.5 | 2.3 | 2.8 | -39 | -56 | -38 |
| Mainz | -4.4 | -5.6 | -5.1 | 3.0 | 2.8 | 2.5 | -40 | -60 | -49 |

**4.The word "improved" is used many times in the paper, not only for the final NO$_2$ result, but also for the retrieval aspects like cloud parameters, albedo, a-priori. To my opinion one should be a bit careful with this term. One can claim something is improved if there is a better match with independent observations, or if obvious biases are removed. This is not always clear. The main conclusion of the paper is an increase of the NO$_2$ columns which better agrees with MAXDOAS (smaller bias).**

We have rephrased the word "improved" e.g. to "*new*" in line 24 and 500, "*updated*" in line 405, and "*corrected*" in line 299, "*higher*" in line 327.

**5.The uncertainty estimates for the individual aspects of the retrieval: cloud fraction/pressure, albedo, profiles could be discussed in more detail. It seems the authors make use of numbers from De Smedt et al, instead of deriving typical uncertainties for GE-LER, ROCINN, POLYPHEMUS. It would be good to have some rough estimates of uncertainty reductions for the individual terms in the new versus old retrieval.**

We have updated the uncertainty estimates in pages 26-27 as *"The estimated parameter uncertainties considered in the AMF uncertainty budget include 0.0016 for GE_LER albedo (Loyola et al., 2020b), 75 hPa for POLYPHEMUS/DLR profile height (estimated from the profile height standard deviation), 0.05 for the OCRA cloud fraction, and 50 hPa ROCINN_CAL cloud pressure (Loyola et al., 2020a). "*

We have included a Table 3 in page 25 showing the individual changes of each step (as suggested by the other reviewer) and the uncertainty reduction.

*Table 3. Main settings for the step-by-step improvements and results of the tropospheric $NO_2$ retrievals for the different steps of the updates for Munich (48.15N, 11.57E). The tropospheric $NO_2$ columns (VCDtrop) are given in absolute values (molec/cm$^2$), and the percentage numbers in the brackets are changes relative to the reference case. The uncertainties in the tropospheric $NO_2$ columns (VCDtropErr) are given in relative values.*

| | Sect. 2.3 (Reference) | Sect. 4.1 | Sect. 4.2 | Sect. 4.3.1 | Sect. 4.3.2 (Improved) |
|---|---|---|---|---|---|
| Surface albedo | OMI LER climatology | TROPOMI GE_LER data | TROPOMI GE_LER data | TROPOMI GE_LER data | TROPOMI GE_LER data |
| A priori $NO_2$ profile | TM5-MP | TM5-MP | POLYPHEMUS/DLR | POLYPHEMUS/DLR | POLYPHEMUS/DLR |
| Cloud parameter | OCRA/ROCINN_CRB version 1.x | OCRA/ROCINN_CRB version 1.x | OCRA/ROCINN_CRB version 1.x | OCRA/ROCINN_CRB version 2.1 | OCRA/ROCINN_CAL version 2.1 |
| VCDtrop (Feb. 2019) | $1.02 \times 10^{16}$ | $1.07 \times 10^{16}$ (+4.9%) | $1.51 \times 10^{16}$ (+48.0%) | $1.57 \times 10^{16}$ (+53.9%) | $1.43 \times 10^{16}$ (+40.2%) |
| VCDtropErr (Feb. 2019) | 65% | 62% | 57% | 56% | 53% |
| VCDtrop (Aug. 2019) | $3.86 \times 10^{15}$ | $4.03 \times 10^{15}$ (+4.4%) | $5.02 \times 10^{16}$ (+30.1%) | $5.13 \times 10^{16}$ (+32.9%) | $4.59 \times 10^{16}$ (+18.9%) |
| VCDtropErr (Aug. 2019) | 41% | 41% | 37% | 36% | 35% |

**6.In our experience with the operational product the impact of the free troposphere is not negligible. This may be discussed in more detail.**

Comparing to the operational data assimilation technique that applies actual meteorological fields, one general limitation of modified reference sector methods is the possible misinterpretation of tropospheric background column over clean regions as stratospheric column. To overcome this issue, (D)STREAM makes use of cloudy pixels with medium altitudes, which directly reflect the actual stratospheric column as the tropospheric column is mostly shielded. However, the broad-scale free tropospheric diffuse $NO_2$ cannot be fully distinguished from stratospheric $NO_2$ by (D)STREAM. We have included more discussions in page 7 regarding the tropospheric background as *"Due to the use of cloudy pixels, which directly reflect the actual stratospheric column as the tropospheric column is mostly shielded,*

*STREAM reduces the bias caused by the free-tropospheric contamination or tropospheric background in the reference region, in comparison with other modified reference sector methods (Beirle et al., 2016). However, the broad-scale free tropospheric diffuse NO₂ might not be fully separated from the stratospheric NO₂, which is a general limitation of the modified reference sector methods."*

**7.The paper mentions that the datasets are available upon request. A brief description of the product would be useful, e.g. are the input fields like GE-LER and OCRA/ROCINN cloud parameters included? Are averaging kernels included?**

We have included an introduction of the product file in Data Availability as "*The DLR TROPOMI NO₂ product are published as HDF version 5 files. For each ground pixel, the TROPOMI data product provides the retrieval results (e.g., the slant column, stratospheric column, and tropospheric column of NO₂), input information (such as the GE_LER surface albedo, POLYPHEMUS/DLR a priori NO₂ profiles, and OCRA/ROCINN cloud parameters), uncertainty estimate, processing quality flag, and averaging kernel.* "

**8.And finally, it would be nice if the authors can provide some recommendations for the future development of the (operational and scientific) TROPOMI NO₂ retrievals in the conclusions section!**

We have extended the future works in page 35 regarding the POLYPHEMUS/DLR a priori profile, GE_LER surface albedo, and MA-DOAS validation.

**Detailed remarks:**

**Title: Improved compared to what? The retrieval and reference should be clearly defined.**

We have updated the title and texts accordingly. Please refer to major comment 1.

**l12: Uncertainty strat column = 3.5 10^14 "for polluted conditions". This is a bit strange: the stratosphere does not have polluted conditions.**

We have updated the expression in line 12 as "$3.5 \times 10^{14}$ molec/cm² *in case of significant tropospheric sources*".

**l25: Decrease from 55 to 34 %: what is the 55% reference?**

We have clarified the reference in line 26 as "The implementation of the algorithm improvements leads to a decrease of the relative difference from -55.3% to -34.7% on average *in comparison with the DLR reference retrieval*."

**l25: At the end of section 6 it is shown that the comparison with MAXDOAS is affected significantly by the differences in the sensitivity profiles for MAXDOAS compared to TROPOMI. With kernel smoothing the remaining difference is about -20%. It would be interesting to mention this.**

We have included the texts in line 26 as "*When the satellite averaging kernels are used to remove the contribution of a priori profile shape, the relative difference decrease further to ~-20%.*"

**l65: There is no reference to the data assimilation approach. It would be useful to explain a bit more this alternative approach.**

We have included references in line 67 as "*(Eskes et al., 2003; Dirksen et al., 2011)*" and extended the description in line 69 as "*Advantages of the data assimilation approach include a realistic error estimation and the capture of small-scale dynamical and chemical variability of stratospheric $NO_2$.*"

**l120: Improvements compared to what? To the previous DLR algorithm or compared to the operational algorithm, or both? Even at the end of the introduction it is still somewhat unclear which two retrievals are compared.**

We have updated the sentence in 125 as "a number of improvements to the tropospheric $NO_2$ retrieval over Europe are introduced *for the DLR product*."

**l151: An intensity offset correction is used, while this is not done in the operational retrieval. Would be good to have a brief discussion of the impact of this intensity offset term. How do the slant columns compare with the operational algorithm?**

Though the precise physical origin of such an intensity offset has not been fully discussed, the intensity offset correction is included in the DLR product to reduce the fitting residues. Figure R5. illustrates the effect of applying a linear intensity offset correction, which decreases the $NO_2$ columns by up to 11% and the fitting residues by up to 23% mainly over the cloud-free ocean, likely due to the compensation of inelastic vibrational Raman scattering in water bodies (Vountas et al., 2003).

[Figure]

Figure R5. Difference in NO₂ columns (slant columns scaled by geometric AMFs) and fitting residues (retrieval root mean square, RMS) estimated with and without a linear intensity offset correction (with-without) on 5 February 2019.

Figure R6 compares the DLR and KNMI NO₂ slant columns, with similar spatial distribution to Fig. R5 mainly reflecting the differences of the intensity offset correction. The results are in agreement with van Geffen et al., 2020. We have referred to *van Geffen et al., 2020* for further discussion in line 158.

[Figure]

NO₂ column density difference [$10^{14}$ molec/cm²] (DLR improved – KNMI operational)

Figure R6. Difference in NO₂ columns (slant columns scaled by geometric AMFs) from the DLR improved algorithm and KNMI operational product (DLR-KNMI) on 5 February 2019.

**Sec 2.2 STREAM: How does STREAM distinguish the stratospheric background from a free tropospheric background? Please add some discussion on the free troposphere, which is supposed to be included in the tropospheric NO₂ column.**

Please refer to major comment 6.

**l182: "average bias of 1e13 molec/cm2 with respect to the ground-based zenith-scattered light differential optical absorption spectroscopy (ZSL-DOAS) measurements" This is a very small number. Please provide the uncertainty range on this comparison. STREAM produces a somewhat larger column than the assimilation approach because it does not distinguish stratosphere from free troposphere.**

We have included the uncertainty in line 193 as "an average bias of $1\pm8\times10^{13}$ molec/cm²".

**Sec 3.1, description of DSTREAM: I am wondering if there is any interference between DSTREAM and the destriping? The directional part removes east-west**

**biases. Does the destriping do something similar? Is destriping done before STREAM is calculated, or after?**

The interference between DSTREAM and the de-striping is expected to be small.

We first have corrected the mistake in the label of Fig. 3 in page 11 as "The de-striping is  implemented here." The correct figure without de-striping is shown in Fig. R7, which shows similar impact of local time changes across the orbit as the results with de-striping.

[Figure]

Figure R7. Similar as Fig. 3 in the manuscript but for measurements without de-striping correction.

[Figure]

Figure R8. NO$_2$ slant columns (scaled by geometric AMFs) averaged for clean regions between 20°S and 20°N with and without de-striping correction for orbit 6748 on 1 February 2019.

The de-striping correction is implemented before the (D)STREAM. Figure R8 shows the effect of applying the empirical de-striping correction. According to the box-car averaging method (Boersma et al., 2011), the total NO$_2$ columns from 60 adjacent viewing angles are averaged for every TROPOMI along-track array. In comparison to the operational de-striping relying on CTM, only relatively small-scale variation is removed, and the larger-scale directional part may not be captured.

**l 257: Why is this latitude weighting introduced? Even though the diurnal cycle effect is smaller at the equator, I would assume it could still be modelled with STREAM? For instance, average slopes with viewing angle could be accomodated as a function of latitude also near the equator.**

[Figure]

Figure R9. NO₂ columns (slant columns scaled by geometric AMFs) for the eastern, central, and western segments of the orbit swath.

The main consideration of the latitude weighting is to reduce interpolation errors for low latitudes with less orbital overlap, as shown in Fig. R9 and explained in line 267.

**l273: Can the difference with the model (3.5e14) be considered a true uncertainty estimate? Or is it a lower limit, e.g. because of the finite model resolution?**

3.5e14 can be regarded as a lower limit due to the use of 0.75 degree resolution for the synthetic data.

**l 290: Does the GE-LER approach also provide an uncertainty estimate?**

In the near future an independent Neural Network will be trained and implemented for error estimation. We have updated the conclusion accordingly.

**l290: How sensitive is the GE-LER to L1B calibration errors? How has the GE-LER been validated, e.g has it been compared with MODIS-based BRDF results? Please**

**add some more info for the reader to judge the performance of the GE-LER approach.**

Large L1B calibration errors will significantly affect the GE_LER retrieval. For UV fitting window, like $O_3$, $SO_2$, and HCHO, the GE_LER data has been validated with representative products, such as OMI and GOME-2 LER climatologies. A first comparison of $NO_2$ GE_LER data has shown a general good agreement with S5P/KNMI and GOME-2 LER climatology, particularly for oceans and the polluted continents, as shown in Fig. R10 for example. We are currently composing a paper about an extensive validation of the GE_LER products.

[Figure]

Figure R10. S5P/KNMI LER climatology at 460 nm and GE_LER retrieved from TROPOMI data (February 2018-2020) in the $NO_2$ window.

**l315: "The surface LER values from GE_LER are lower than the climatological OMI values by 0.03 on average" Is this a statement for February or August? It seems that the average differences in August are smaller than in February (by looking at the figures).**

We have corrected the statement in line 331 as "*The mean differences between the surface LER values from GE_LER and OMI climatological values are lower than 0.03.*"

**Fig. 7: Please comment on the very low albedo values over the Mediterranean compared to the OMI LER.**

The very low albedo values could be due to errors in the input parameters, such as SCD or the constant term of DOAS polynomial. As compared to the OMI LER based on years of data, the GE_LER used in this study considered a short time period of data, as explained in line 334. We have included the expression in line 335 as "which makes GE_LER more likely affected by aerosol contamination *or outliers in the input data*". Further analyses are required to explain the low values.

**Sec 4.2: Please provide the definition of the POLYPHEMUS European domain (lat-lon domain boundaries).**

We have included the domain in line 346 as "for Europe *(34.0N-60.4N, 12.0W-40.2E)* in this study".

**l339: "European TNO-MACC". Please specify the version and reference year.**

We have included the version and reference year in line 356 as "European TNO-MACC *(from 2011 with base year 2005)*".

**l349: "The profile shape from POLYPHEMUS/DLR agree better with the MAX-DOAS measurements". The profiling capabilities of the MAXDOAS instruments are limited and often also quite uncertain. Part of this profile shape is a-priori defined. So, does it make sense to compare these profiles?**

We have removed the comparison of profiles and updated the expression in line 367 as "*In comparison with the tropospheric AMF calculated using the MAX-DOAS $NO_2$ profile, the bias reduces from 0.32 (48.5%) for TM5-MP to -0.04 (-6.0%) for POLYPHEMUS/DLR.*"

**Fig.10: It would be nice to also show a direct comparison of POLYPHEMUS tropospheric $NO_2$ column against TROPOMI (using the averaging kernels), e.g. also for February and August. Would it be possible to add these plots?**

We have included the plot in pages 19-20. Please refer to major comment 2.

**Sec. 4.2: Has POLYPHEMUS been compared to the CAMS regional modelling results? If so, please provide a brief summary of these comparisons. Please provide more detail on how the model has been validated in general, e.g. against observations of $NO_2$.**

Please refer to the major comment 2 for the comparison between POLYPHEMUS/DLR and EUROS-LOTOS as well as the validation of POLYPHEMUS/DLR.

**Sec. 4.2: Fig.9 indicates that POLYPHEMUS has basically very low concentrations of $NO_2$ in the free troposphere. Even though the free tropospheric column is small, this may have quite a significant impact on the AMF, and may in part be a reason why we see only red colors in Fig. 10. With larger free tropospheric NOx concentrations I would expect more blue colors especially in the more remote areas away from the main pollution hotspots. Please comment on this.**

From Fig 9 left panel, POLYPHEMUS/DLR shows larger free tropospheric $NO_2$ concentration. Please refer to major comment 2 for further discussions.

**Fig.11: "comparisons with version 1.x " Please be more explicit. Do you show cf(2.1)-cf(1.x) or cf(1.x)-cf(2.1)? What is x in "1.x" in this case?**

We have updated the plot in page 21 with text "*(version 2.1 – version 1.x)*". We have added the description "*(1.0 and 1.1)*" in line 402.

**Fig.12: Why do you show a normalised CF instead of a mean CF?**

We have changed the Fig. 13 in page 22 to show the mean CF instead of a normalized one.

**Is Fig.12 consistent with Figs. 11 and 13 and the text? Fig.12 indicates smaller CF at the right side of the orbit in v2.1, while e.g. line 389 mentions an increase of the cloud fraction?**

We have updated the plot. Please refer to the previous reply.

**Fig.13: Again: do you show cf(2.1)-cf(1.x) or cf(1.x)-cf(2.1)?**

We have updated the Fig.14 in page 22 with "*(version 2.1 – version 1.x)*".

**Fig.13: Is there any evidence that the new v2.1 cloud pressures are better than the old ones? Please discuss this in more detail.**

The biggest change for ROCINN between v1.x and version 2.1 is the replacement of the coarse MERIS surface albedo climatology used in v1.x with an actual surface albedo retrieval with the matching TROPOMI scene information in real time (GE_LER retrieval). Hence, shortcomings of a climatology (e.g. spatial resolution, short term temporal variations like snow/ice coverage, cloud contamination) are largely reduced by the GE_LER retrieval which is based on TROPOMI data themselves and performed at the same time with the TROPOMI measurements, hence it is more representative of the actual scene conditions than a climatology can be. Although the ROCINN retrieval scheme for cloud height and optical thickness / cloud albedo itself did not change, an improvement for those parameters is achieved indirectly through the improved surface albedo, particularly over scenes with rapidly varying surface conditions like snow/ice.

**Fig.14: How can I understand the reduction in $NO_2$ over the Po Valley? The cloud pressure does not seem to change much here.**

The reduction in NO₂ over the Po Valley is mainly due to a new check in the version 2.1 processor: if the cloud fraction is smaller than 0.1 and the height difference between the retrieved cloud height and the surface height is lower than 100 m, the scene is assumed to be cloud-free. We have included the explanation in page 23 as *"The NO₂ reductions over the Po Valley are related to an additional check implemented in the version 2.1 processor: the pixel is assumed to be cloud-free for the almost clear-sky condition (cloud fraction < 0.1) with the retrieved cloud height very close to the surface height (difference < 100 m). This correction improves the data yield of the TROPOMI cloud products compared with other satellite cloud products; the performance of this correction under different surface conditions (dark, bright, snow, ice) or under presence of different types of low-level aerosols (fog, smoke, dust, ash) is under investigation."*

**l407: Could you also provide the average % difference in the AMF in winter and summer for CAL vs CBR?**

We have included the relative difference as "by more than $1{\times}10^{15}$ molec/cm² *(18%)* for polluted regions in winter" in line 441 and "less than $5{\times}10^{14}$ molec/cm² *(10%)* for summer" in line 443.

**Fig. 16: Please be explicit what is shown: CRB-CAL or CAL-CRB**

We have included the information (CAL cloud -CRB cloud) in the figure in page 25.

**l419: Please also provide the % increase in winter and summer.**

We have included the relative difference in line 454 as "enhanced by $2{\times}10^{15}$ molec/cm² *(37%)* in winter and $8{\times}10^{14}$ molec/cm² *(15%)* in summer".

**Figures 8, 10, 16 provide the contributions to Fig. 18. However, it would be good to also have the numbers for the contributions of the various terms (new STREAM, new profiles, GE-LER, CAL) to the increase. Could such numbers be provided for February and August?**

We have included the individual changes of each step in Table 3. Please refer to major comment 5.

**l427: I think it is good to refer to van Geffen 2020 here, who discuss this in detail. The 4.5e14 is similar/close to the 5.2e15 estimated for the operational retrieval.**

Done.

**l433: The table in the paper of De Smedt is mentioned. Are these numbers used without any modification? Are these consistent with estimates for e.g. the GE-LER**

and OCRA/ROCINN CAL uncertainties? I suggest to include the relevant numbers in the text! Some more discussion on the uncertainties related to GE-LER, OCRA and ROCINN would be very relevant. For clear sky the AMF uncertainty is estimated as 20%.

we have updated the uncertainty estimation. Please refer to major comment 5.

**l469: "which is mostly explained by the relatively low sensitivity of spaceborne measurements near the surface, the aerosol shielding effect, and the gradient smoothing effect." This is not so clear. The retrieval accounts for the lower sensitivity at the surface, and with the new GE-LER these uncertainties are hopefully reduced. The aerosol shielding effect was discussed as implicitly accounted for via the cloud retrieval. So it is not clear if a bias should remain due to these effects.**

We have reformulated the expression in page 30 as "The $NO_2$ levels are underestimated by 34% by TROPOMI with a standard deviation of 16%, *which is likely explained by comparison errors (such as the gradient smoothing effect, the comparison choices, and the inherent difference in sensitivity), partly by* the remaining impact of structural uncertainties in the satellite data (Boersma et al., 2016, such as the impact of the choice of the a priori $NO_2$ profiles and/or the albedo database assumed for the satellite AMF calculations), and by the different measurement types or the specific conditions of the validation sites." Please see also the discussion to the last comment for the conclusion section.

**Table 4: It would be great if also numbers for the operational product could be included.**

We have included the operational validation. Please refer to major comment 3.

**Figure 23: Nice to see that also the operational product is included!**

**Fig. 24: Nice to see this plot! The sensitivity profiles of MAXDOAS and satellite are very different, so it is good to demonstrate the impact on the comparison.**

**l500: "see Sec 5.2" ? Section 5.2 discusses the uncertainties and does not discuss the kernels.**

We have removed the expression in line 542.

**Conclusions section: It would be relevant to comment on the differences between the new DLR retrieval (and inputs) and other $NO_2$ retrieval approaches (operational, NASA, POMINO, ECCC, BEHR, Bremen) and discuss possible recommendations following from this comparison of retrieval methods.**

We have included the comparison in page 35 as "*The TROPOMI NO₂ research product from DLR is a complement to the operational product due to the use of independent approaches for stratosphere-troposphere separation and AMF calculation. Comparing to the other regional TROPOMI NO₂ product, the DLR European retrieval reduces the potential biases introduced by using inputs from different instruments or climatologies and confirms the importance of applying more realistic input parameters with better resolution for AMF calculation.*"

**Conclusions section: The bias in the updated retrieval against MAXDOAS is reduced from 55 to 34%, but is still substantial. Do the authors have an opinion what are the main retrieval aspects causing this difference? How can this gap between surface and satellite observations be closed? Something is said about this in section 6, but it would be interesting to discuss it again in the conclusions, and perhaps including recommendations for a way foreward.**

Part of the remaining biases are likely related to the spatial heterogeneity effect (e.g. Pinardi 2020; Goldberg 2019), limiting the comparison of a satellite pixel (even if smaller for TROPOMI than what we were used to before) to ground-based data having a few-to-some Km length line-of-sight sensibility. Tang et al., 2021 calculated the sub-grid variability from airborne campaign measurements over Korea and Los Angeles basin, finding between 15 and 20% impact for TROPOMI pixel size. Illustration of this effect around Uccle site, based on TROPOMI data itself, is discussed in Sect. 6, Fig 22, with impact up to 19% in summer.

We have included more recommendations in page 36 as "*Further improvements in the ground-based validation include using the full MAXDOAS line-of-sight sensitivity and the intersect with the TROPOMI pixel(s) or having more ground-based instruments located within a TROPOMI pixel. More frequent ground-based measurements and measurements in more than one direction might better sample the temporal and spatial variability around the measurement sites (Richter and Lange, 2021; Dimitropopulou et al., 2020).*"

[revised manuscript text omitted]

---

## Author Comment (AC2)

**Response to Referee #2**

We thank the reviewer for the careful reading and the useful comments. Below we give the reviewer's comment, our response, and the changed text in the manuscript. The page and line numbers refer to the mark-up version of the manuscript.

1. **It is a long paper (43 pages) and involves so many models, making it pretty difficult to read thru the whole paper. So it will be very helpful if the authors could provide an error budget table which will highlight the biggest contributors to the overall accuracy improvement of the retrieval system compared to the reference models. That way readers can easily see the relative importance of one model/input parameter to others, even though Fig. 19 and section 7 (conclusions) provide some detailed numbers about uncertainty and improvement.**

We have included a Table 3 in page 25 showing the individual changes of each step and the uncertainty reduction.

*Table 3. Main settings for the step-by-step improvements and results of the tropospheric $NO_2$ retrievals for the different steps of the updates for Munich (48.15N, 11.57E). The tropospheric $NO_2$ columns (VCDtrop) are given in absolute values (molec/cm$^2$), and the percentage numbers in the brackets are changes relative to the reference case. The uncertainties in the tropospheric $NO_2$ columns (VCDtropErr) are given in relative values.*

| | Sect. 2.3 (Reference) | Sect. 4.1 | Sect. 4.2 | Sect. 4.3.1 | Sect. 4.3.2 (Improved) |
|---|---|---|---|---|---|
| Surface albedo | OMI LER climatology | TROPOMI GE_LER data | TROPOMI GE_LER data | TROPOMI GE_LER data | TROPOMI GE_LER data |
| A priori $NO_2$ profile | TM5-MP | TM5-MP | POLYPHEMUS/DLR | POLYPHEMUS/DLR | POLYPHEMUS/DLR |
| Cloud parameter | OCRA/ROCINN_CRB version 1.x | OCRA/ROCINN_CRB version 1.x | OCRA/ROCINN_CRB version 1.x | OCRA/ROCINN_CRB version 2.1 | OCRA/ROCINN_CAL version 2.1 |
| VCDtrop (Feb. 2019) | $1.02 \times 10^{16}$ | $1.07 \times 10^{16}$ (+4.9%) | $1.51 \times 10^{16}$ (+48.0%) | $1.57 \times 10^{16}$ (+53.9%) | $1.43 \times 10^{16}$ (+40.2%) |
| VCDtropErr (Feb. 2019) | 65% | 62% | 57% | 56% | 53% |
| VCDtrop (Aug. 2019) | $3.86 \times 10^{15}$ | $4.03 \times 10^{15}$ (+4.4%) | $5.02 \times 10^{16}$ (+30.1%) | $5.13 \times 10^{16}$ (+32.9%) | $4.59 \times 10^{16}$ (+18.9%) |
| VCDtropErr (Aug. 2019) | 41% | 41% | 37% | 36% | 35% |

2. **Sensitivity analysis and case studies have been made for almost all important input parameters. However, readers may wonder when doing sensitivity analysis or case study of one parameter, what the values of other input**

**parameters are. The authors can improve on this by listing the default values of all input parameters. That is, it does not change until it becomes the parameter of analysis.**

We have included the list of default values. Please refer to major comment 1.

3. **There are a few places which have inconsistency issues. For example, L157-163, p6 x-track striping issue is introduced and de-striping approach is discussed. However, it is not clear if de-striping correction has been applied to the slant column retrieval (not mentioned thereafter). There is discussion on how to remove VZA dependence of the stratospheric NO2 columns in DSTREAM as shown in Fig. 6 by dividing TROPOMI orbit swath into 3 segments (western, central and eastern), which, however, is not the approach discussed in page 6, L157-163 for de-striping.**

[Figure]

Figure R1. NO₂ slant columns (scaled by geometric AMFs) averaged for clean regions between 20°S and 20°N with and without de-striping correction for orbit 6748 on 1 February 2019.

The de-striping correction is applied to the slant columns to reduce the non-physical across-track variation, as shown in Figure R1. According to the box-car averaging method (Boersma et al., 2011), the total NO₂ columns from 60 adjacent viewing angles are averaged for every TROPOMI along-track array. The method removes relatively small-scale variation, and the larger-scale directional part may not be captured.

The directional dependency correction for DSTREAM is applied during stratospheric NO₂ estimation to consider the diurnal variation of stratospheric NO₂ across the track, as illustrated in Fig. 6 in the manuscript.

We have clarified that the de-striping correction is applied to the slant columns in line 166 as "To reduce the systematic stripes, a de-striping correction *is applied to the TROPOMI NO₂ slant columns, which* is calculated…"

**Also, AMF is calculated using TROPOMI GE_LER data for surface albedo and OCRA/ROCINN_CAL model for cloud parameter as described in Table 1. But in SCD calculation (Eq.7), cloud pressure is from ROCINN_CRB model (see L302).**

We have removed the inconsistency in line 316 as "the use of the effective *scene* pressure $p_e$, *which is provided in the new version 2.1 processor for OCRA/ROCINN (see Sect. 4.3.1)*".

**The box-AMFs in Eq.(3) tabled values are calculated using VLIDORT (what is the version number?), while radiance quantities for cloud radiance fraction computation (see Eq.5) and SCD calculation (see Eq.7) are simulated using LIDORT (again, which version?). As we know, different versions of LIDORT and VLIDORT may produce inconsistent simulation results.**

The VLIDORT version 2.7 is used in Eq. (3), and the LIDORT version 3.6 is used in Eqs. (5) and (7). From Table R1, consistent settings are implemented for VLIDORT v2.7 and LIDORT v3.6. We have included the version information accordingly in the manuscript (lines 205, 225, and 314,).

Table R1. Major features of LIDORT and VLIDORT. Table adapted from Spurr et al., 2015.

| Feature | LIDORT Version | VLIDORT Version |
|---|---|---|
| Pseudo-spherical (solar beam attenuation) | 2.1 | 1.0 |
| [Enhanced spherical (line-of-sight)] | 2.2+ | 2.1 |
| Green's function treatment | 2.3 | n/a |
| 3-kernel BRDF + linearization | 2.4 | 2.2 |
| Multiple solar zenith angles | 3.0 | 2.2 |
| Solution saving, BVP telescoping | 3.0 | 2.3 |
| Linearized thermal & surface emission | 3.2 | 2.4 |
| Outgoing sphericity correction | 3.2 | 2.3 |
| Total Column Jacobian facility | 3.3 | 2.4 |
| Transmittance-only thermal mode | 3.3 | 2.4 |
| Fortran 90 release | 3.5 | 2.5 |
| BRDF supplement | 3.5 | 2.5 |
| Structured I/O | 3.5 | 2.5 |
| External SS | 3.6 | 2.6 |
| BRDF upgrade and surface-leaving supplements | 3.6 | 2.6 |
| Atmospheric and surface blackbody Jacobians | 3.7 | 2.7 |
| Codes made thread-safe for parallel computing | 3.7 | 2.7 |
| Introduction of Taylor series expansions | 3.7 | 2.7 |
| BRDF and surface-leaving supplement upgrades | 3.7 | 2.7 |

4. **Some other minor issues**
    1. **Another cloud parameters retrieval algorithm – FRESCO is mentioned in L226-229, p9-10. But it seems that FRESCO is not used in this study, so why it is discussed here?**

The FRESCO cloud algorithm is introduced due to the importance to the operational KNMI NO2 product. As we have added more discussions as well as validation results for the operational data (as suggested by the other reviewer), a brief introduction of the methods used in the operational product may help to understand the differences.

2. **It is suggested to change "The IFS(CB05BASCOE) model" to "The IFS (Integrated Forecast System) model" in L262, p11, and remove "Integrated Forecast System" in the following line (L263). Also, in L267, change "using IFS(CB05BASCOE) data" to "using CB05BASCOE data".**

We have reformulated the sentence in line 275 as "*Particularly advantageous for stratospheric studies, the tropospheric chemistry module in the Integrated Forecast system (IFS) is extended* with the stratospheric chemistry" and removed "using IFS(CB05BASCOE) data".

3. **Fig.11, right panel, why does cloud fraction have negative values down to -0.2?**

We have changed the Fig. 13 in page 22 to show the mean CF instead of a normalized one.

4. **L371, how can ROCINN retrieve effective cloud pressure and cloud albedo values at cloud fraction of 0?**

We have removed the misleading expression and rephrased the texts in line 398 as "*the effective scene pressure and effective scene albedo values, which are added in the version 2.1 processor,* are applied…"

5. **L395, it implies that the multiple scattering between the cloud bottom and the ground is not considered in the CRB cloud model. That simply is not true.**

We have removed the sentence in line 429.

6. **Fig.16, right panel, there is not much one can see. Suggested to change the color bar scale (reduce the up limit) to enhance the red color.**

We have update the colorbar on the right panel.

**7. In section 6, page 25, why Uccle was selected as an example showing in Figs. 20-22. Readers may wonder why this site (not other site) was selected.**

Uccle is selected as an example due to its suburban location, which is suitable to illustrate the gradient smoothing effect, as introduced in Fig. 23 in page 31.

**References**

[revised manuscript text omitted]

---

## Author Response (AR1)

**Response to Referee Henk Eskes**

We thank the reviewer Henk Eskes for the careful reading and the usefulmments. Below we give the reviewer's comment, our response, and the changed text in the manuscript. The page and line numbers refer to the mark-up version of the manuscript.

**General remarks:**

1. The reader may find the title and abstract confusing. It suggests that the authors are improving the operational TROPOMI retrieval product, while actually the paper discusses and upgrade of the retrieval product implemented at DLR. This should be made more clear at the top of the abstract and in the title. I suggest the authors make clear which retrieval product they refer to, for instance by calling it the DLR TROPOMI NO2 European product. Maybe good to add the word "scientific" as opposed to the "operational" retrieval.

We have updated the title to "An improved TROPOMI tropospheric NO2 research product over Europe" and referred the data to "the DLR TROPOMI NO2 European product" throughout the paper.

2.The preformance of the POLYPHEMUS model is not easy to judge. The resolution of this model is not very high, 0.2x0.3 degree, while TROPOMI has a resolution of 0.05 degree. The CAMS models for instance run at 0.1x0.1 degree. This resolution is in between the TM5-MP and TROPOMI. Would higher resolution produce a major further improvement? I would like to see a comparison between POLYPHEMUS and TROPOMI as extra figure. The POLYPHEMUS model seems to produce very low free tropospheric NO2 concentrations. Is this expected to impact the retrieval?

The 0.2x0.3 degree is selected for POLYPHEMUS/DLR due to the feasibility in terms of runtime and affordability in terms of computing resources. We have analysed the effect of model resolution using a priori NO2 profiles from LOTOS-EUROS, which constitutes one of the state-of-the-art atmospheric chemistry models used by the regional Copernicus Atmosphere Monitoring Service (CAMS, http://www.copernicus-atmosphere.eu). Figure R1 shows the differences in the tropospheric NO2 columns retrieved by altering the model resolution for LOTOS-EUROS (0.1x0.1 degree - 0.2x0.3 degree) in February 2019. The increase of the spatial resolution from 0.2x0.3 to 0.1x0.1 degree improves the tropospheric NO2 columns moderately by up to  $5 \times 10^{14}$  molec/cm2 or 11% for polluted regions.

Figure R1. Differences in the tropospheric NO2 columns retrieved by altering the model resolutions (0.1x0.1 degree - 0.2x0.3 degree) for LOTOS-EUROS a priori NO2 profiles in February 2019.

We have included a new Fig. 11 in page 20 comparing the tropospheric NO2 columns from TROPOMI measurements and POLYPHEMUS/DLR simulations (using the satellite averaging kernel) and included the description in pages 19 as "Figure 11 compares the tropospheric NO2 columns from TROPOMI measurements and POLYPHEMUS/DLR simulations. The satellite averaging kernel, which describes the vertical sensitivity of measurements of NO2 concentrations, is applied to reduce the systematic biases caused by unrealistic a priori profile information. From Fig. 11, POLYPHEMUS/DLR NO2 columns are generally higher than satellite measurements, which can be partly related to the use of the TNO-MACC emission dataset (Denier van der Gon et al., 2010). An update of the POLYPHEMUS/DLR model using the more recent TNO-MACC\_II emission (Kuenen et al., 2014) is planned for the near future. We note here that the profile shape is of far more importance than the column bias for the interpretation of satellite retrievals."

---

## Author Response (AR2)

**Response to Referee**

We thank the reviewer for the suggestions. Below we give the reviewer's comment, our response, and the changed text in the manuscript.

**General remarks:**

**1.** I was happy to see the extra figure 11 added, showing Polyphemus vs. Tropomi. However to my opinion figures R1, R2, R3 and R4 are also relevant to the reader. I would suggest that R2, R3 and R4 are added in the supplement, and that the the resolution effect (R1) is summarised in 1-2 sentences in the main text, section 4.2.

We have included R2, R3, and R4 in the supplement, and included the texts in Sect. 4.2 as "Sections S1 and S2 in the supplement compare the POLYPHEMUS/DLR NO2 results with data from ground-based stations as well as the regional chemistry transfer model LOTOS-EUROS, indicating that POLYPHEMUS/DLR is in general reliable." We have included the discussion of resolution (R1) as "An additional increase of the spatial resolution from 0.2x0.3 to 0.1x0.1 degree affects the tropospheric NO2 columns moderately by up to  $5 \times 10^{14}$  molec/cm2 or 11% for polluted regions." in Sect. 4.2.

2. (Fig. 9) There is clearly a big difference in the VMR of TM5-MP and Polyphemus between 800 and 400 hPa. To my opinion TM5-MP may be more realistic here. Regional model often neglect processes like lightning, aircraft emissions or deep convection which are of importance for the free/higher troposphere. This impacts the overall profile shape. Please comment on this and the possible implication for the retrieval.

To avoid confusion we would like to address that Fig. 9b shows the normalized profiles (instead of the original profiles), as indicated in the texts, which show large differences between 800 and 400 hPa. We agree with the reviewer that the global TM5-MP model considers additional processes such as lightning, with enhanced NOx injected into the upper troposphere (Williams et al., 2017). We have calculated the tropospheric AMF with POLYPHEMUS/DLR NO2 concentrations at 400-800hPa increasing from ~0.02 to ~0.05 (approximately TM5 values) ppb, and the impact on NO2 retrieval is limited (smaller than 1%) for the example in Fig 9.

3. The emissions in the model are outdated. Using reported negative trends in NO2 (e.g. EEA air quality reports), how much of the difference in Fig. 11 could be explained by this?

Figure R1 shows the tropospheric  $NO_2$  columns from POLYPHEMUS/DLR simulations with the TNO-MACC emissions from the year 2011 and 2018. The update of emissions reduces the overestimations of tropospheric  $NO_2$  columns by up to 50%, which likely

explains ~50% of the difference in Fig. 11. We have included the texts in Sect. 4.2 as "An update of the POLYPHEMUS/DLR model using the more recent TNO-MACC\_II emission is planned for the near future, which reduces the overestimations of tropospheric  $NO_2$  columns by up to 50%."

Figure R1. Tropospheric NO2 columns from POLYPHEMUS/DLR simulations with the TNO-MACC emissions from the year 2011 (top) and 2018 (bottom) with the TNO-MACC emissions from the year 2011 (top) and 2018 (bottom) over Europe in July 2018.

4. Concerning my question about "I am wondering if there is any interference between DSTREAM and the destriping?" It was good to see the answer from the authors and Figs R7/R8. However, I would like to see a few sentences in the paper, section 3, to summarise this. Also my question "Can the difference with the model (3.5e14) be considered a true uncertainty estimate?" Also here it would be good to see a modification in the paper text, section 3.2.

We have included the texts in Sect. 3 as "The de-striping correction is implemented before STREAM, but only small-scale variation is removed.". We have included the texts in Sect. 5.2 as "(*the values can be regarded as a lower limit due to the use of the finite resolution for the synthetic data*)".

**References**

Williams, J. E., Boersma, K. F., Le Sager, P., and Verstraeten, W. W.: The high-resolution version of TM5-MP for optimized satellite retrievals: description and validation, Geosci. Model Dev., 10, 721–750, https://doi.org/10.5194/gmd-10-721-2017, 2017.